# An evolutionarily-conserved Wnt3/β-catenin/Sp5 feedback loop restricts head organizer activity in *Hydra*

Matthias C. Vogg[1], Leonardo Beccari[1], Laura Iglesias Ollé[1], Christine Rampon[2,3,4], Sophie Vriz[2,3,4], Chrystelle Perruchoud[1], Yvan Wenger[1] & Brigitte Galliot[1]

Polyps of the cnidarian *Hydra* maintain their adult anatomy through two developmental organizers, the head organizer located apically and the foot organizer basally. The head organizer is made of two antagonistic cross-reacting components, an activator, driving apical differentiation and an inhibitor, preventing ectopic head formation. Here we characterize the head inhibitor by comparing planarian genes down-regulated when *β-catenin* is silenced to *Hydra* genes displaying a graded apical-to-basal expression and an up-regulation during head regeneration. We identify Sp5 as a transcription factor that fulfills the head inhibitor properties: leading to a robust multiheaded phenotype when knocked-down in *Hydra*, acting as a transcriptional repressor of *Wnt3* and positively regulated by Wnt/β-catenin signaling. *Hydra* and zebrafish Sp5 repress *Wnt3* promoter activity while *Hydra Sp5* also activates its own expression, likely via β-catenin/TCF interaction. This work identifies Sp5 as a potent feedback loop inhibitor of Wnt/β-catenin signaling, a function conserved across eumetazoan evolution.

[1] Department of Genetics and Evolution, Institute of Genetics and Genomics in Geneva (iGE3), Faculty of Sciences, University of Geneva, 30 Quai Ernest Ansermet, CH-1211 Geneva 4, Switzerland. [2] Centre Interdisciplinaire de Recherche en Biologie (CIRB), CNRS UMR 7241/INSERM U1050/Collège de France, 11, Place Marcelin Berthelot, 75231 Paris Cedex 05, France. [3] Université Paris Diderot, Sorbonne Paris Cité, Biology Department, 75205 Paris Cedex 13, France. [4] PSL Research University, 75005 Paris, France. Correspondence and requests for materials should be addressed to B.G. (email: brigitte.galliot@unige.ch)

The freshwater *Hydra* polyp, which belongs to Cnidaria, a sister group to Bilateria, has the remarkable talent to regenerate any lost body parts, including a fully functional head. *Hydra*, which is made of two cell layers, external named epidermis and internal named gastrodermis, shows a polarized tubular anatomy with a head at the apical/oral pole and a foot at the basal/aboral one, both extremities being enriched in nerve cells (Fig. 1a). Remarkably, the cnidarian oral pole has been proposed to correspond to the posterior end of bilaterians[1]. Head regeneration relies on the rapid transformation of a piece of somatic adult tissue, the amputated gastric tube, into a tissue with developmental properties named head organizer, which directs the patterning of the regenerating tissue (reviewed in[2–4]) (Fig. 1b). This process is highly robust in *Hydra*, occurring after bisection at any level along the body column. The concept of organizer was first discovered by Ethel Browne who performed lateral transplantation experiments between pigmented and depigmented *Hydra*[5]. By grafting a non-pigmented piece of head onto the body column of a pigmented host, she observed the development of an ectopic axis predominantly made of pigmented cells, demonstrating the recruitment of host cells by the graft. This discovery was later confirmed in hydrozoans[6–10] but also in vertebrates where organizers play an essential role during embryonic development[11]. In *Hydra* regenerating its head, the organizer gets established within 10 to 12 h after mid-gastric bisection, restricted to the head-regenerating tip within the first 24 h, remaining stable until the new head is formed and subsequently persisting as a homeostatic head organizer[9].

The *Hydra* model also helped understand the dual structure of organizers. By comparing the efficiency of apical grafts to induce ectopic axis on intact or decapitated hosts, Rand et al. showed that the *Hydra* head organizer exerts two opposite activities, one activator that promotes apical differentiation, and another inhibitory that prevents the formation of supernumerary or ectopic heads[12]. In *Hydra* the inhibitory activity is graded along the body axis, maximal at the apical pole[8], and tightly modulated during head regeneration, rapidly decaying after amputation and slowly recovering[13]. Gierer and Meinhardt used the results obtained from a series of transplantation experiments to propose a general mathematical model of morphogenesis[14]. Their model revisits the Turing model based on the reaction-diffusion model, where two substances that exhibit distinct diffusion properties and interact with each other, form a minimal regulatory loop that suffices for de novo pattern formation[15]. Gierer and Meinhardt posed that the activation component acts over short-range distance and the inhibition one over long-range distance. They distinguished between "*the effective concentrations of activator and inhibitor, on one hand, and the density of their sources on the other*"[14]. These models proved to efficiently simulate basic properties of pattern formation and to fit molecular data in a variety of developmental contexts[16].

In *Hydra*, the Holstein lab identified *Wnt3* as a growth factor fulfilling the criteria of the head activator, expressed locally at the tip of the head in intact *Hydra*, rapidly re-expressed in head-regenerating tips after amputation, and able to trigger an autocatalytic feedback loop[17–19]. Concerning the head inhibitor necessary to maintain a single head in homeostatic polyps and to develop a single head in budding and regenerating contexts, several attempts were made to characterize it, either biochemically or genetically. A protease-resistant small hydrophilic molecule was identified, exhibiting an apical to basal graded activity although with some activity also detected in the basal disc[20,21]. This last property discouraged from any further characterization. A genetic screen identified a *Hydra* ortholog of the vertebrate Wnt dickkopf inhibitors, named *hyDkk1/2/4*, which efficiently antagonizes Wnt activity in *Xenopus*[22]. However, *Dkk1/2/4* is not

expressed apically, being negatively regulated by Wnt/β-catenin signaling and its downregulation does not induce a multiheaded phenotype[22,23]. A recent study suggests that *Hydra* Thrombospondin might be involved in head inhibition, however its downregulation does not lead to a multiheaded phenotype[24]. Therefore, the molecular nature of the negative regulator(s) of the *Hydra* head organizer remains unknown. Here we used a strategy based on the evolutionarily conservation of Wnt/β-catenin signaling to trace the *Hydra* head inhibitor. We identify the transcription factor Sp5 as a transcriptional repressor of Wnt3, leading to a robust multiheaded phenotype when knocked-down in *Hydra*, while Wnt/β-catenin signaling positively modulates *Sp5* expression. Sp5 fulfills the requirements of a head inhibitor in *Hydra*, and we show that this feedback loop between Sp5 and Wnt/β-catenin signaling appears conserved across eumetazoan evolution.

## Results

**Identification of putative *Hydra* head inhibitors**. To identify inhibitors of apical patterning that regulate the activity of the head organizer in both homeostatic and regenerative conditions, we established five criteria to be fulfilled by head inhibitor (HI) gene(s): (1) be controlled by Wnt/β-catenin signaling, (2) display an apical-to-basal graded activity, (3) be upregulated within the first day of head regeneration, (4) inhibit Wnt/β-catenin signaling, (5) prevent head formation (Fig. 1b). To select β-catenin target genes, we used a dataset of 440 genes downregulated in planarians silenced for *β-catenin*[25] to retrieve 124 *Hydra* cognate genes (Supplementary Data 1). We analyzed their spatial and temporal RNA-seq expression profiles and found 5/124 genes predominantly expressed in the head and 3/5 upregulated in head-regenerating tips at least 1.5 fold after 24 h of regeneration (Fig. 1c, d). Among these candidates, we found *Wnt3* and *Wnt5*, known as positive regulators of morphogenetic processes[17,18,26] and *Sp5*, previously identified as a Wnt/β-catenin target gene in vertebrates[27–31], thus a putative HI candidate (Fig. 1e). *Hydra Sp5* (*HySp5*) encodes a Sp/Klf-class transcription factor whose sequence clusters with the bilaterian Sp5 ones in phylogenetic analyses (Supplementary Figs. 1–3).

Whole mount in situ hybridization confirmed the RNA-seq *Sp5* pattern in intact *Hydra*, predominantly expressed in the head although absent from the apical tip where *Wnt3* expression is maximal (Fig. 1f, g). After mid-gastric bisection, *Sp5* is rapidly upregulated in both head- and foot-regenerating tips but its expression is only sustained in head-regenerating ones (Fig. 1g, Supplementary Fig. 4) supporting the idea that *Sp5* is involved in head but not foot regeneration. We also performed a RNA-seq analysis of the cell-type expression[32] and found that both *Sp5* and *Wnt3* are predominantly expressed in the gastrodermal epithelial stem cells (gESCs), a cell type associated with morphogenetic processes (Supplementary Fig. 5).

**Hydra Sp5 a robust head inhibitory component**. Next, we silenced *Sp5* by electroporating siRNAs in intact animals and observed that within two days following the third electroporation (RNAi3), *Sp5*(RNAi) animals develop ectopic axes, initially from the budding zone, few days later from the upper body column (Fig. 2a, Supplementary Fig. 6). These ectopic axes differentiate multiple heads when located in the basal half but not from the upper half. Both ectopic axes and ectopic heads express the apical markers *Wnt3*, *Bra1* and *Tsp1*, and the gland cell marker *Kazal1* in the gastric tissue (Fig. 2b). When single-headed animals silenced for *Sp5* are bisected after RNAi2, they all regenerate multiple heads that express *Wnt3* at the tip (Fig. 2c, Supplementary Fig. 7). This multiheaded phenotype is

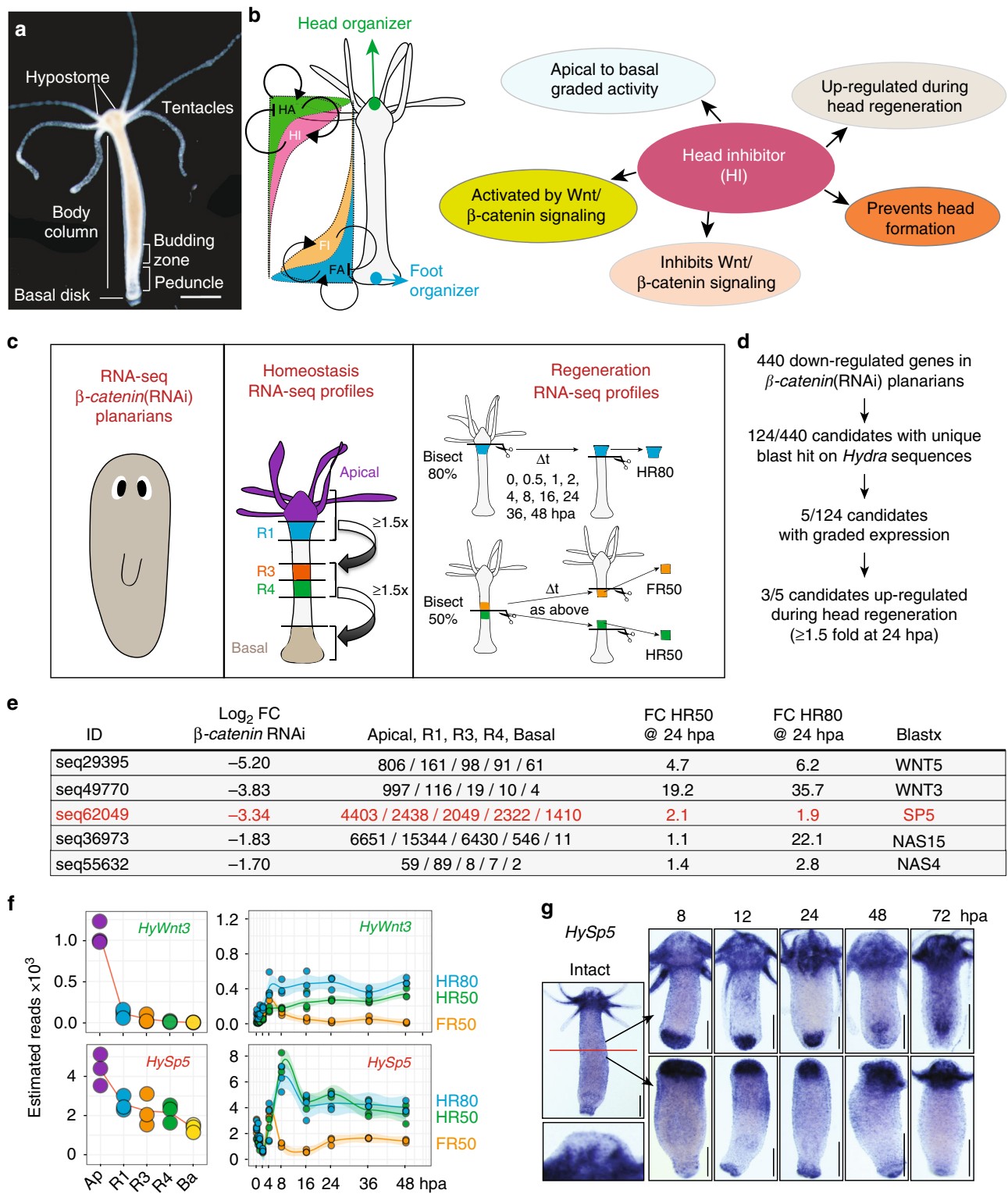

robust, emerging quite synchronously in 50% uncut animals one day after RNAi2, in 100% two days after RNAi3 (Fig. 2d, Supplementary Fig. 6a–c). Furthermore, these ectopic heads express the neuropeptide RF-amide and are able to catch and ingest live *Artemia*, indicating that each ectopic head is functional (Fig. 2e, Supplementary Fig. 6d, Supplementary Movies 1–4). These results indicate that Sp5 acts as a strong inhibitor of head formation in *Hydra*.

**Sp5 antagonizes Wnt/β-catenin signaling in *Hydra*.** Next, we tested whether the multiheaded phenotype corresponds to a de-repression of *Wnt3*. To do this, we first tested whether the phenotype occurs when the Wnt/β-catenin pathway is inactive and thus knocked-down *Sp5* together with *β-catenin* (Fig. 3a). Silencing *β-catenin* on its own delays head regeneration (Supplementary Fig. 8) and causes the formation of ectopic bumps in intact animals (Fig. 3a, Supplementary Fig. 9). While knocking

**Fig. 1** Screening strategy to identify candidate head inhibitor genes in *Hydra*. **a** Anatomy of an intact *Hydra*. The apical extremity (head) is composed of a dome-shaped structure called hypostome, surrounded by a ring of tentacles. At the other extremity (foot), the basal disk allows the animals to attach. **b** The five criteria used to identify HI candidate genes. **c, d** Screening procedure applied to identify HI candidate genes: An RNA-seq dataset of 440 downregulated genes in *β-catenin* (RNAi) planarians was used to retrieve through blastx on NCBI (*E* value < 1e$^{-10}$) 124 non-redundant *Hydra* sequences that correspond to 106 unique proteins (Supplementary Data 1). These candidates were next tested on RNA-seq data sets obtained in intact *Hydra* measured at five positions along the body axis (apical -Ap-, regions R1, R3, R4, basal -Ba-) to identify five apical-to-basal graded genes, which were tested on RNA-seq data sets obtained from regenerating tips taken at nine time points after a 50% or 80% bisection. Data available on HydrAtlas.unige.ch **e** Three genes downregulated after *β-catenin*(RNAi) in planarians, show an apical-to-basal graded expression in *Hydra*, and a minimal 1.5-fold upregulation in head-regenerating tips at 24 hpa. The 3rd column indicates the mean value of the number of reads measured in three biological replicates in the indicated regions. Fold Change (FC) measured in head-regenerating (HR) tips at 24 h post-amputation (hpa) over the values measured at time 0. **f** *Wnt3* and *HySp5* RNA-seq profiles in intact and regenerating animals. **g** *HySp5* expression patterns in intact and regenerating *Hydra* tested as indicated after mid-gastric bisection in two independent experiments. Inset: magnified view of the apex. Scale bars: 250 μm

down *Sp5* causes the formation of multiple heads, the simultaneous knockdown of *Sp5* and *β-catenin* prevents the occurrence of the multiheaded phenotype (Fig. 3a, Supplementary Fig. 9), suggesting that an increase in Wnt/β-catenin signaling activity is necessary to trigger multiple head formation when *Sp5* is knocked down.

To further demonstrate that Sp5 represses Wnt/β-catenin signaling via *Wnt3* repression, we knocked-down *Sp5* in combination with Alsterpaullone (ALP), a drug that activates the Wnt/β-catenin pathway by antagonizing GSK3β[33,34]. As anticipated this combination led to a significant increase in ectopic tentacle formation, while knocking down *β-catenin* provides the opposite effect (Fig. 3b, Supplementary Fig. 10a). In these *Sp5*(RNAi) animals, we could also detect an increase in *Wnt3* expression along the body column, indicating that *Sp5* does repress *Wnt3* expression (Fig. 3c, Supplementary Fig. 10b).

We also performed reaggregation experiments with cells coming from ALP-treated animals knocked-down either for *Sp5* or for *β-catenin*. In standard conditions of reaggregation, several head spots form, each of them containing 5–15 *Wnt3* expressing cells at 24 hours[35]. When *Sp5* is knocked-down, we noted that the reaggregates tend to form multiple axes with a number of *Wnt3* expressing spots increased by two-fold (Fig. 3d, e, Supplementary Fig. 11). In contrast, when *β-catenin* is knocked-down, the reaggregation process proceeds slower with aggregates exhibiting only few tentacles at day-4, with a number of *Wnt3*-expressing clusters similar to that observed in scramble(RNAi) control animals (Fig. 3d). These results confirm that Sp5 directly or indirectly represses *Wnt3* expression.

To test whether Sp5 can directly repress the *Wnt3* promoter, we produced a transgenic strain expressing the *HyWnt3*–2149: GFP-*HyAct*:dsRed construct where 2'149 bp of the *Hydra Wnt3* promoter drives GFP expression and the *Hydra Actin* promoter drives dsRed expression[19]. We noted distinct levels of *Wnt3*-driven GFP fluorescence in control transgenic animals, maximal at the apex, intermediate in the adjacent region above the tentacle ring, and null at the level of the tentacle ring and along the body column (Supplementary Fig. 12a). In such transgenic animals knocked-down for *Sp5*, we did not record any body-wide GFP fluorescence but rather the appearance of patches of GFP + cells at the tip of the ectopic axes (Supplementary Fig. 12b). We could confirm this patchy *Wnt3* activation along the body column of *Sp5*(RNAi) animals by performing a detailed kinetic analysis of *Wnt3* expression (Supplementary Fig. 12c–d). This *Wnt3* ectopic expression pattern suggests that *Sp5* is silenced in restricted regions along the body column where *Wnt3* is de-repressed and enhances its own expression via β-catenin signaling as previously recorded.

**Sp5 represses the *Hydra* and zebrafish *Wnt3* promoter**. To further investigate the repressing activity of HySp5 on the

*HyWnt3* promoter, we performed luciferase reporter assays in human HEK293T cells (Fig. 4a–c). As the *HyWnt3*–2149:Luc construct shows a very low basal activity, we co-expressed a constitutively active form of β-Catenin (*CMV*:huΔβ-Cat)[36] that enhances by ~10-fold the luciferase activity (Fig. 4b). In such conditions, the co-expression of HySp5 significantly reduces the activity of the *HyWnt3* promoter (Fig. 4b). This effect was not observed when a partial version of HySp5 lacking the DNA-binding domain was used, indicating that the repressive effect of HySp5 is DNA-binding dependent (Fig. 4b). Two adjacent cis-regulatory modules were previously identified in the *HyWnt3* promoter, a 599 bp-long activator that contains three clustered TCF binding sites and a 386 bp-long repressor sequence[19], located immediately downstream (Fig. 4a, Supplementary Fig. 13a). This repressor module, highly conserved across *Hydra* species (Fig. 4a), is necessary for the Sp5-mediated *Wnt3* repression, as the repression is no longer observed when this element is removed (Fig. 4b). Among the four constructs that harbor limited deletions within the *Wnt3* repressor element, the construct containing both the -386/-286 and the -95/-1 sequences is the only one repressed by Sp5 (Fig. 4c), suggesting that the Sp5-dependent *Wnt3* repression requires the cooperative activity of these two elements.

To test whether Sp5 also represses *Wnt3* transcription in vertebrates we tested the 4 kb promoter region of the zebrafish *Wnt3* locus in reporter assays where the zebrafish paralogs ZfSp5a and ZfSp5l1 are expressed (Fig. 4d, e). As for the *HyWnt3*–2149 construct, the transcriptional activity of the *ZfWnt3*–3997 construct was strongly enhanced by huΔβ-Cat, but repressed upon co-expression of ZfSp5a or ZfSp5l1 (Fig. 4e). The repressor activity of ZfSp5a was abolished when the DNA-binding domain was deleted. Although the zebrafish *Wnt3* promoter does not share obvious sequence homologies with that of the *HyWnt3* promoter, we could identify regions evolutionarily-conserved across different teleost lineages as well as TCF binding sites (TCF-BS) (Fig. 4d, Supplementary Fig. 13b). ChIP-qPCR experiments performed in transfected HEK293T identified two evolutionarily-conserved elements within the *ZfWnt3* promoter directly bound by ZfSp5a (Fig. 4f).

**Wnt/β-catenin signaling regulates *HySp5* expression**. In planarians as in zebrafish and mammals, the canonical Wnt/β-catenin pathway positively regulates the expression of *Sp5*[25,27,31,37]. In mammals, Sp5 has also been reported to auto-regulate its expression, although studies in human and mouse embryonic stem cells (ESCs) differ on whether Sp5 acts positively or negatively on its own promoter[31,37]. In *Hydra*, a two days exposure to ALP suffices to upregulate *Sp5* expression along the body column (Fig. 5a, Supplementary Fig. 14), suggesting that *Sp5* regulation by the Wnt/β-catenin pathway predates the divergence of cnidarians. To test this hypothesis, we cloned 2'992 bp of the

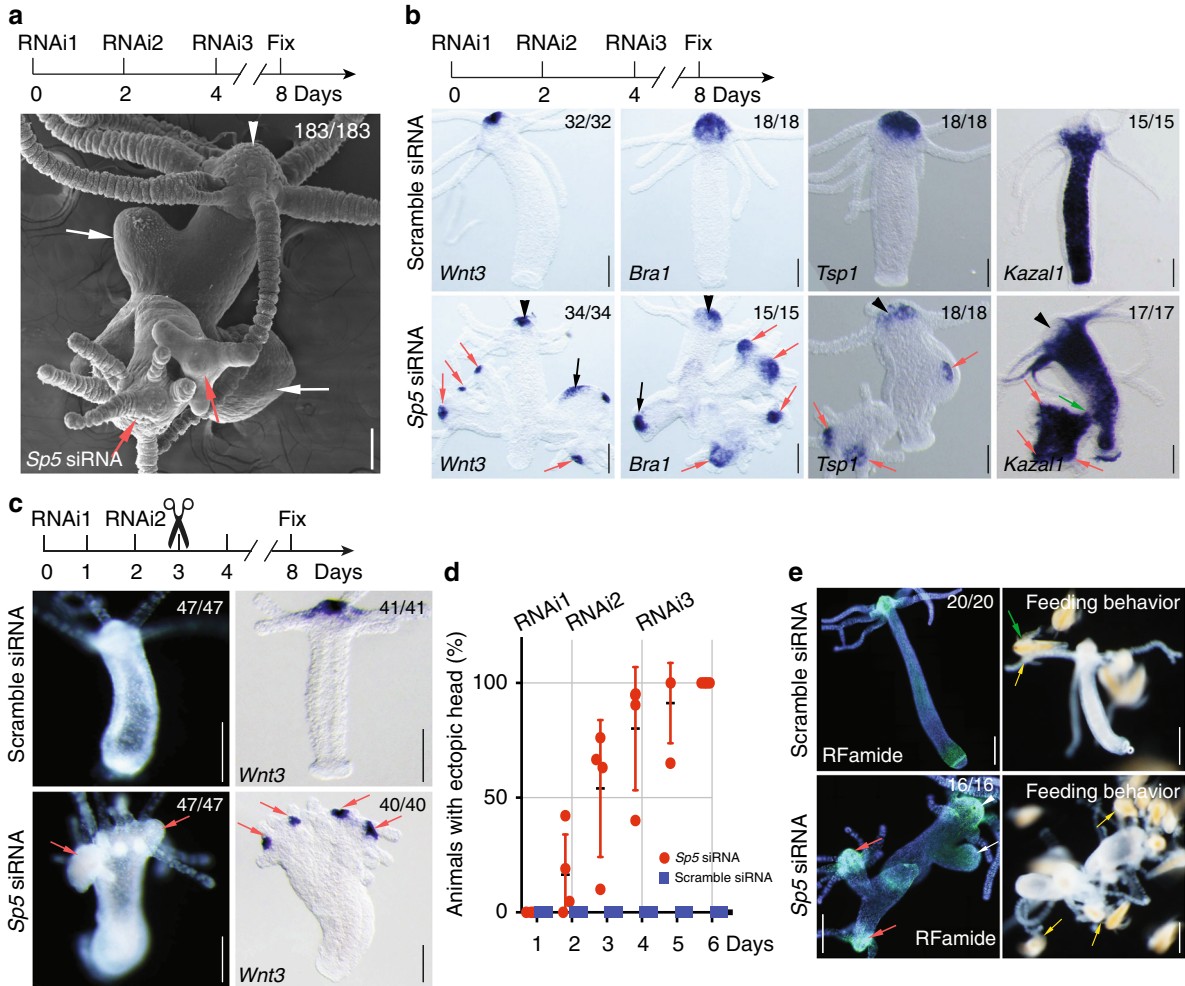

**Fig. 2** Knocking down *HySp5* leads to the formation of ectopic axes and ectopic heads. **a**, **b** Morphological changes detected in intact *Hydra* exposed three times to *Sp5* or scramble siRNAs as indicated. In **a** SEM view four days after *Sp5* RNAi3. In **b** animals fixed and detected for *Wnt3*, *HyBra1*, *Tsp1* or *Kazal1* expression. White or black arrows: ectopic axes; red arrows: ectopic heads; arrowheads: parental heads; green arrow: connection between endodermal tissue of the secondary axis and the parental body column (3 independent experiments). **c** Morphological changes detected in head regenerating animals 5 days after mid-gastric bisection performed 24 h after two exposures to *Sp5* siRNAs (3 independent experiments). Arrows as in **a**. **d** Kinetics of morphological changes observed in intact *Sp5*(RNAi) *Hydra*. Each dot represents an independent experiment (*n* = 4). **e** RFamide expression and feeding response tested in ectopic heads of *Sp5*(RNAi) *Hydra* 4 days after RNAi3 (2 independent experiments). White arrow: ectopic axis; red arrows: ectopic heads; arrowhead: parental head; green arrow: *Artemia*; yellow arrows: *Artemia* caught by tentacles. Scale bars: 200 μm. Error bars indicate SD

*HySp5* promoter, a region that is evolutionarily-conserved across *Hydra* species and contains five putative TCF binding sites (Fig. 5b, Supplementary Fig. 13c). We evidenced its responsiveness to Wnt/β-catenin signaling, as we recorded a significant upregulation of the activity of the *HySp5*–2992:Luc reporter construct when the human WNT3, LRP6 or huΔβ-Cat proteins were co-expressed (Fig. 5c). In addition, we found that HySp5 can bind its own promoter as in ChIP-qPCR experiments Sp5 binding is significantly enriched in two neighboring regions located immediately upstream of the Sp5 Transcriptional Start Site (TSS) (Fig. 5d). Furthermore, co-expression of *HySp5*–2992:Luc and HySp5, alone or in combination with huΔβ-Cat resulted in a strong increase in luciferase activity (Fig. 5e). In mouse ESCs, Sp5 interacts with β-catenin and Tcf-Lef1 to regulate gene expression[31]. As anticipated, we found in a ChIP-seq analysis the mouse Sp5 and β-catenin proteins enriched in the same region of the Sp5 promoter (Supplementary Fig. 15a), suggesting a possible cooperation to regulate *Sp5* transcription. We performed co-immunoprecipitation experiments with HEK293T cells co-transfected with HySp5 and huΔβ-Cat or huTCF1 and

observed an interaction between HySp5 and these factors (Fig. 5f, Supplementary Fig. 15b–c). These results indicate that HySp5, similarly to its mammalian cognates, can act as an activator or a repressor of transcription and that *Hydra* and vertebrate Sp5 can interact with β-catenin or TCF1.

**Sp5 DNA-binding properties are evolutionarily-conserved**. To further compare the transcriptional activities of HySp5 and ZfSp5a, we expressed HySp5 or ZfSp5a in HEK293T cells and analyzed the genomic occupancies and the transcriptional changes induced by their overexpression (Fig. 6a). ChIP-seq analysis revealed that HySp5 binds a much smaller fraction of sequences than ZfSp5a (Fig. 6b), while the number of genes bound by HySp5 and ZfSp5a is not so different, 13'251 vs. 18'619, 99% of the HySp5 bound genes are also ZfSp5a targets (Fig. 6c). Interestingly, HySp5 and ZfSp5a differ in the spatial distribution of their target sequences: the majority of HySp5 bound elements localize within the 5 kb proximal region of the assigned genes, while ZfSp5a proportionally binds more frequently elements

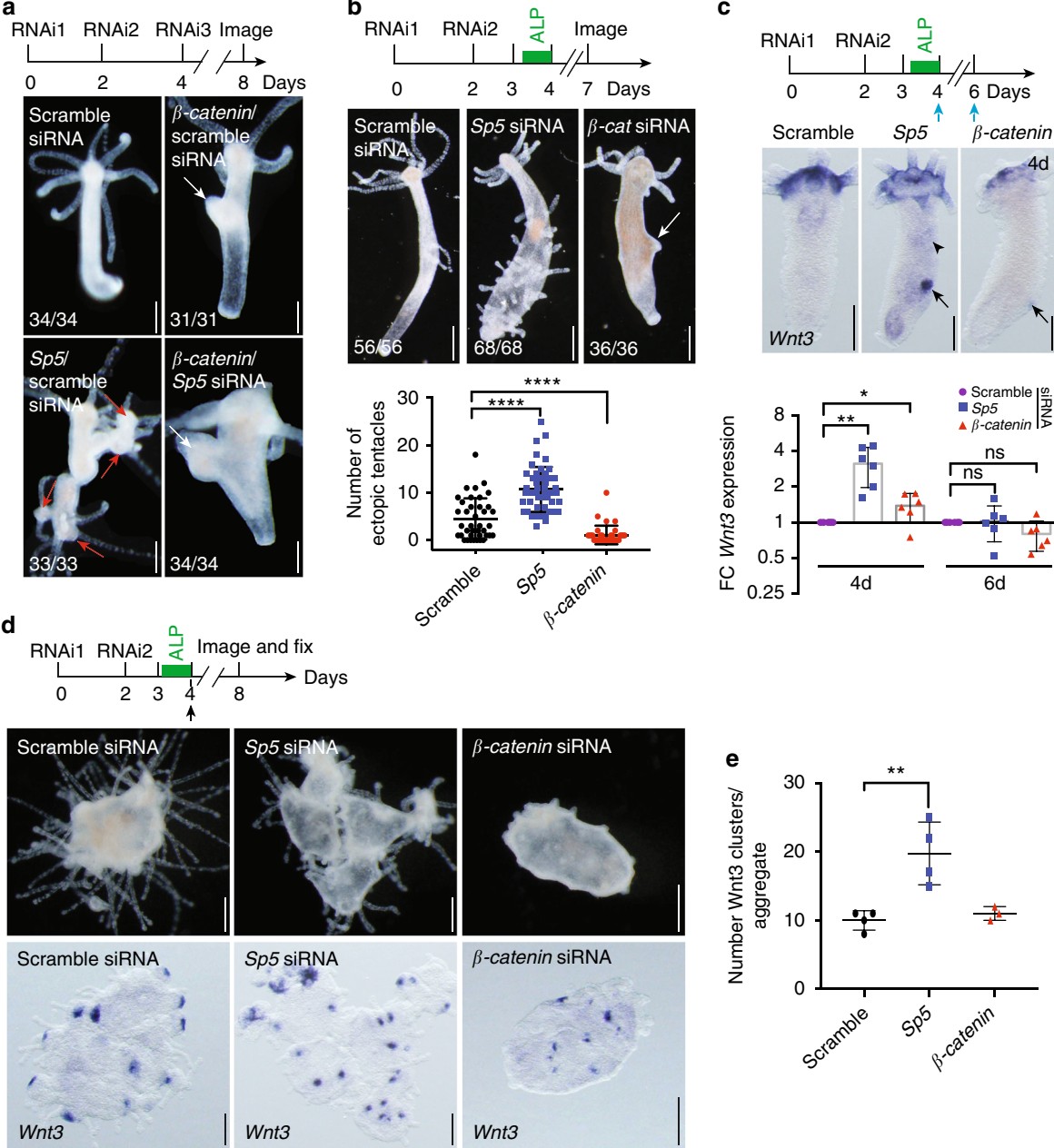

**Fig. 3** HySp5 antagonizes Wnt/β-catenin signaling in *Hydra*. **a** Double knockdown of *Sp5* and *β-catenin*. Animals were pictured live on day-8 in two independent experiments. RNAi: siRNA electroporation day. Red arrows: ectopic heads; white arrows: bumps. **b** Knockdown of *Sp5* and *β-catenin* in ALP-treated animals. Upper panel: Morphological changes observed in animals fixed at day-7. Lower panel: Quantification of ectopic tentacles formed at day-7. Each dot represents the number of ectopic tentacles in one animal (two biologically independent experiments). **c** Intact *Hydra* exposed to siRNAs and ALP as indicated and detected for *Wnt3* by WISH (upper panel) and qPCR (lower panel). Blue arrows: RNA extraction days; black arrows: local increase in *Wnt3*; black arrowhead: diffuse increase in *Wnt3* expression. Each data point represents one biologically independent experiment. **d** Effect of *Sp5* knockdown on head patterning during the process of reaggregation initiated on day-4 (black arrow). Aggregates were pictured live (upper row), then fixed to detect *Wnt3* expression (lower row) in two independent experiments. **e** Quantification of *Wnt3* expressing clusters. Each dot represents the number of *Wnt3*+ clusters in one aggregate. Statistical *p* values: *$P \leq 0.05$, **$P \leq 0.01$, ****$P \leq 0.0001$ (unpaired *t* test). Scale bars in panels **a**–**d**: 200 μm. Error bars indicate SD

located in upstream sequences, above 10 kb from the TSS (Fig. 6d, e). This suggests that vertebrate Sp5 more readily recognizes sequences enriched in long-range regulatory elements, which are not recognized by the HySp5 protein.

Motif enrichment analysis of the HySp5 and ZfSp5a bound elements revealed that the two orthologs recognize both similar and divergent consensus binding sites (Fig. 6f). In both cases, the most enriched motif resembled the general SP/KLF consensus sequence (GGGxGGG/A). We then used the enriched motifs to identify putative HySp5/ZfSp5a binding sites in the regulatory regions of *HyWnt3*, *ZfWnt3* and *HySp5*. We could identify putative HySp5 binding sites in the two regions of the *HyWnt3* repressor required to inhibit transcription (Supplementary Figs. 13). Similarly, we also found evolutionarily-conserved Sp5 binding sites in the regions of *ZfWnt3* and *HySp5* enriched in the ChIP-qPCR analysis, supporting the idea that *Hydra* and

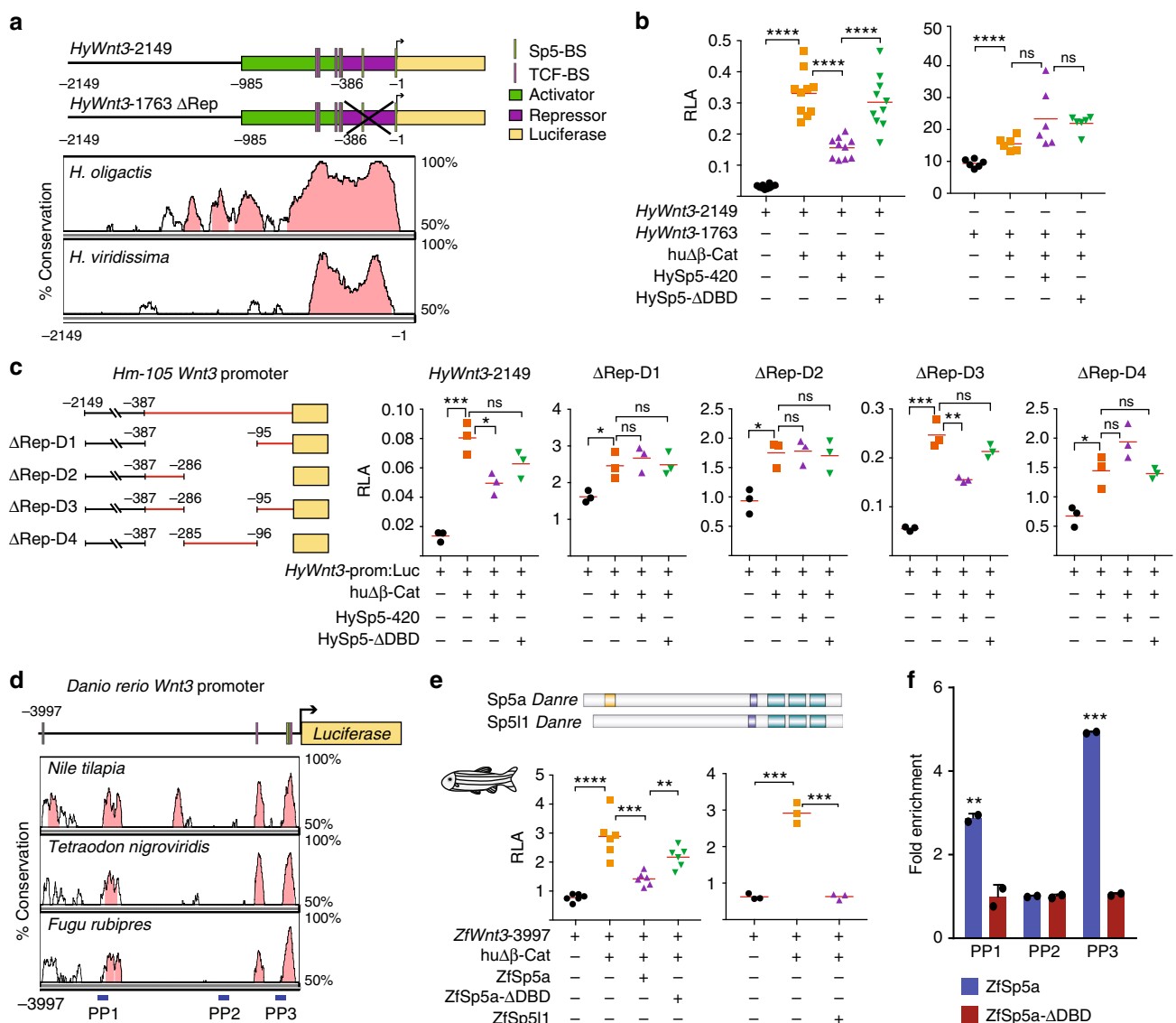

**Fig. 4** *Hydra* Sp5 and zebrafish Sp5 repress the *Wnt3* promoter activity. **a** Map of the *HyWnt3* promoter and phylogenetic footprinting plot comparing the 2 kb genomic region encompassing the *H. magnipapillata (Hm-105) Wnt3* promoter with the corresponding regions in the *H. oligactis* and *H. viridissima* genomes. Green and magenta bars indicate the activator and repressor regions identified by (ref. [19]). Conserved TCF binding sites (TCF-BS) are shown in magenta and putative Sp5-BS in green. Pink peaks in the Vista alignment plot represent evolutionarily-conserved modules (at least 70% base-pair identity over 100 bp sliding window). **b** Luciferase assays measuring the activity of the *HyWnt3*–2149 (left) or *HyWnt3*–1763-ΔRep (right) promoters in HEK293T cells co-expressing a constitutively active human β-Catenin (huΔβ-Cat), HySp5–420 (full-length Sp5) or HySp5-ΔDBD (Sp5 lacking the DNA-Binding Domain). Note the ~300x higher basal activity of *HyWnt3*–1763-ΔRep:Luc when compared to that of *HyWnt3*–2149:Luc. RLA: Relative Luciferase Activity. **c** Luciferase assays performed in HEK293T cells testing the *HyWnt3* promoter when deleted of different portions of the repressor. Note that the repressive effect of HySp5 is only observed with the *HyWnt3*–2149 and *HyWnt3*-ΔRep-D3 constructs. **d** Phylogenetic footprinting plot comparing the 4 kb genomic region encompassing the zebrafish *Wnt3* promoter with the corresponding genomic regions of three teleost fish species. Pink peaks as in panel a; blue rectangles indicate regions of the *ZfWnt3* promoter tested for ZfSp5a binding in ChIP-qPCR assays. PP: Primer Pair. **e** Luciferase assays measuring the activity of the zebrafish *Wnt3* promoter in HEK293T cells, co-transfected with huΔβ-Cat, ZfSp5a, ZfSp5a-ΔDBD (left) or huΔβ-Cat, ZfSp5l1 (right). **f** ChIP-qPCR assays performed with cells expressing ZfSp5a or ZfSp5a-ΔDBD. Note the significant enrichment in the PP1 and PP3 regions. Source Data are provided as a Source Data file. Each data point in **b**, **c**, **e**, **f** represents one biological independent experiment. Statistical *p* values: *$p \leq 0.05$; **$p \leq 0.01$; ***$p \leq 0.001$; ****$p \leq 0.0001$ (unpaired *t* test). Error bars indicate SD

zebrafish Sp5 directly regulate the transcriptional activity of these promoters. Despite the similarity in the main consensus sites bound by HySp5 and ZfSp5a, we also identified motifs differentially enriched among the elements bound by these two orthologs (Fig. 6f). Interestingly, ZfSp5a binds elements that display an over-representation of Tbx1 and Sox13 motifs, which were not identified in the pool of HySp5 bound sequences (Fig. 6f). Members of the Tbx and Sox families are known to

interact with Sp1[38] and β-catenin[39] respectively, suggesting that they could also form transcriptional complexes with Sp5. Thus, the enrichment in Tbx/Sox consensus sequences suggests that vertebrate Sp5 but not *Hydra* Sp5 may regulate gene expression in complexes involving these transcription factors.

To further validate that HySp5 has similar DNA-binding properties than its vertebrate orthologs, we inspected the HySp5 genomic coverages in the proximities of genes identified as

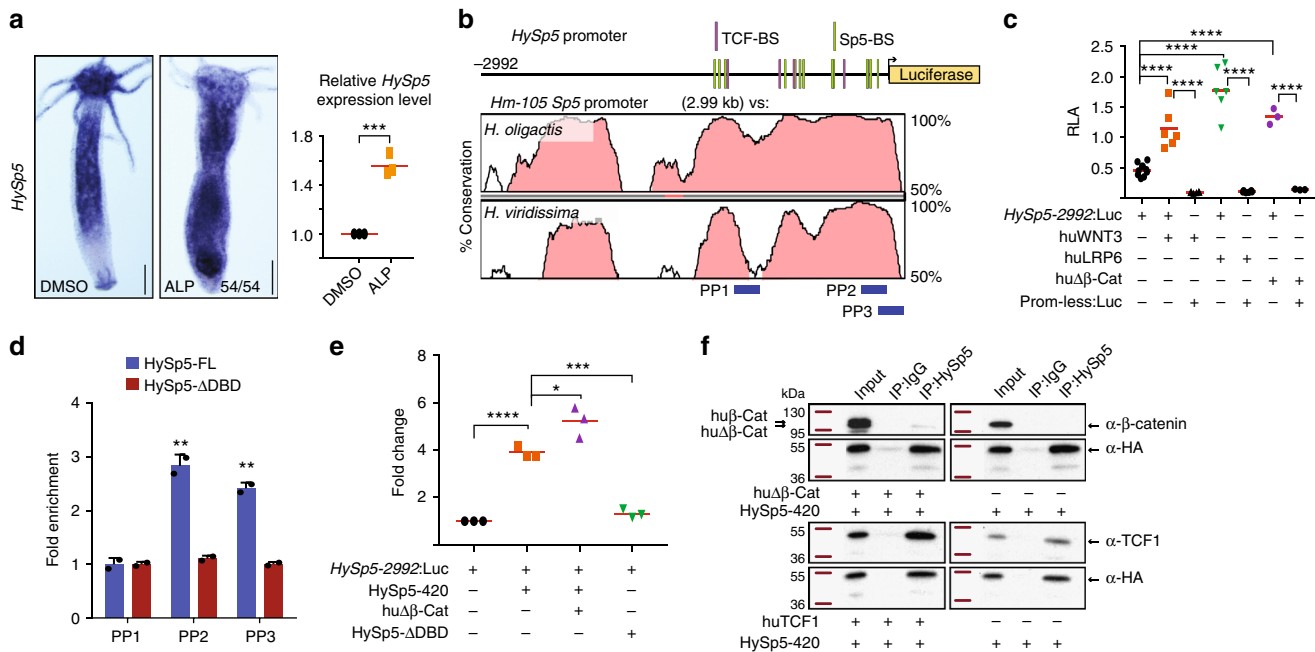

**Fig. 5** Wnt/β-catenin signaling regulates *HySp5* expression. **a** *HySp5* expression in intact *Hydra* exposed to ALP for two days, detected by WISH (left) (3 independent experiments) and qPCR (right). Each point represents an independent replicate. Scale bar: 250 μm. **b** Map of the 2'992 bp genomic region encompassing the *Sp5* promoter from the *Hm-105* strain and phylogenetic footprinting plot comparing this region in *Hm-105*, *H. oligactis* and *H. viridissima*. Pink peaks and binding sites as in Fig. 4. Blue rectangles: regions tested for Sp5 binding in ChIP-qPCR assays. **c** *HySp5* promoter activity measured in HEK293T cells co-transfected with *HySp5*–2992:Luc and plasmids expressing activators of Wnt/β-catenin signaling, human WNT3, LRP6, Δβ-Catenin. **d** ChIP-qPCR assays performed in HEK293T cells expressing *HySp5*–2992:Luc together with HySp5–420 or HySp5-ΔDBD. Source Data are provided as a Source Data file. **e** *HySp5* promoter activity measured in HEK293T cells expressing *HySp5*–2992:Luc with huΔβ-Catenin, HySp5–420 or HySp5-ΔDBD. **f** Immunoprecipitation (IP) of HA-tagged HySp5–420 expressed in HEK293T cells together or not with huΔβ-Catenin (upper) or huTCF1 (lower). IP was performed with an anti-HA antibody and Co-IP products were detected with the anti-β-catenin or anti-TCF1 antibodies. Same results were obtained in two independent experiments. Each data point in (c-e) represents one biological independent experiment. Statistical *p* values: *\*p ≤ 0.05, \*\*p ≤ 0.01, \*\*\*p ≤ 0.001, \*\*\*\*p ≤ 0.0001 (unpaired *t* test). Error bars indicate SD. PP primer pair, RLA relative luciferase activity

Wnt/β-catenin targets in mouse and human ESCs[31,37]. Comparable binding profiles of HySp5 and mSp5 were observed for the *Axin2*, *Bra* and *Lrg5* loci in human and mouse cells, while quite different at the *Nanog* and *Plk4* loci, the latter likely due to cell-type or species specific differences (Supplementary Fig. 16a). We also found a strong enrichment of HySp5 and ZfSp5a binding in the *WNT3* intronic sequences, in the promoter and intronic sequences of the neighboring *WNT9B* locus and in the upstream and intronic sequences of *SP5* (Supplementary Fig. 16b). The GO term enrichment analysis actually identified the Wnt pathway as the most enriched category (Supplementary Fig. 16c, Supplementary Data 2).

All together, these results point to similar DNA-binding capacities between HySp5 and ZfSp5a even though the latter recognizes a larger set of sequences, often located at mid-long distances upstream from the TSS, possibly acting in combination with Sox and/or Tbx proteins.

**Conserved and divergent transcriptional functions of Sp5.** To assess the transcriptional activity of HySp5 and ZfSp5a, we measured by qRNA-seq the transcriptional changes induced by the overexpression of HySp5 and ZfSp5a in HEK293T cells co-expressing or not the huΔβ-Cat construct (Fig. 6a). As controls we used HEK293T cells transfected with a mock plasmid, the huΔβ-Cat construct alone or the mutated HySp5-ΔDBD and ZfSp5a-ΔDBD constructs. Principal component analysis (PCA) showed that HySp5 and ZfSp5a transfected samples, either alone or in combination with huΔβ-Cat, segregated together, widely separated from the control or HySp5-ΔDBD/ZfSp5a-ΔDBD

values (Fig. 6g). This suggests that HySp5 and ZfSp5a elicit overall similar transcriptional responses. Instead, the values obtained from huΔβ-Cat transfected cells grouped together with the values from mock-transfected samples, while the values corresponding to cells co-expressing huΔβ-Cat with HySp5 or ZfSp5a do not substantially differ from those overexpressing HySp5 or ZfSp5a alone (Fig. 6g, Supplementary Data 2). These results imply that HEK293T cells do not respond to huΔβ-Cat overexpression, in agreement with previous reports showing that although HEK293T cells respond to Wnt signalling stimulation by translocating β-catenin to the nucleus[40,41], they display limited transcriptional responses of their endogenous Wnt target genes[37,42].

Next, we analyzed the genes whose expression is modulated upon HySp5 or ZfSp5a overexpression but remains unaffected when their respective DNA-binding domain is deleted (Fig. 6h, Supplementary Data 2). We focused our analysis on the modulated genes that were associated to HySp5- or ZfSp5a-bound elements in ChIP-seq analysis, suggesting that these genes are directly activated or directly repressed targets. We identified downregulated genes, 153 upon HySp5 expression, 113 by ZfSp5a, and 83 by both (Fig. 6i, Supplementary Fig. 17, Supplementary Data 3). This demonstrates that the cnidarian and vertebrate Sp5 proteins have a similar repressive capacity. We also identified 137 and 23 genes upregulated upon ZfSp5a and HySp5 overexpression, respectively. Of these, only 5 are activated by both Sp5 orthologs (Fig. 6i, Supplementary Fig. 17, Supplementary Data 3), indicating that the activator function of the cnidarian and vertebrate Sp5 transcription factors diverged

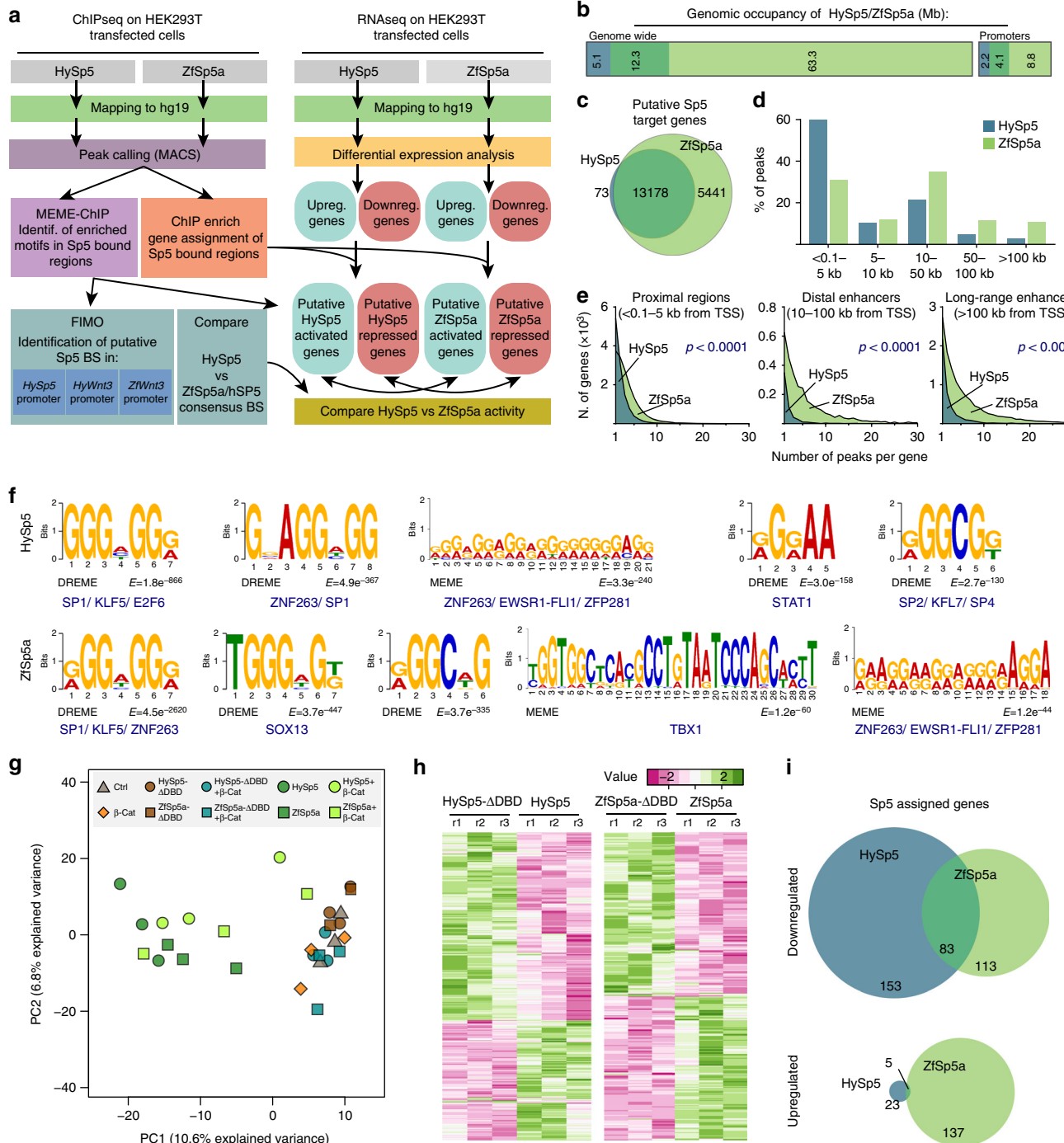

**Fig. 6** *Hydra* Sp5 acts as a transcriptional activator and repressor. **a** Schematic representation of the workflow used for the analysis of the ChIP-seq and RNA-seq data in this study. Putative target genes were identified based on the identification of Sp5 bound elements using ChIPenrich. The consensus motifs enriched in Sp5 bound elements were identified using the MEME ChIP suite tool. Differential expression analysis was performed on RNA-seq samples to identify up- and downregulated genes. Those associated to an Sp5 bound element were considered as direct Sp5 up- or downregulated targets. **b** Bar graph representing the genomic coverage of HySp5 and ZfSp5a genome wide or in the promoters of Ensembl genes (defined as the 5 kb upstream of the gene TSS). Only autosomal chromosomes were used for this study. ZfSp5a coverages are considerably higher than those of HySP5. However, within gene promoters this difference is proportionally lower. **c** Venn diagram representing the number of genes assigned to HySp5 or ZfSp5a bound elements. Note the considerable overlap between HySp5 and ZfSp5a data sets. **d** Bar plot representing the percentage of Sp5 bound elements at different distances from the assigned gene TSS for HySp5 (blue) or ZfSp5a (green). **e** Frequency distribution of the number of peaks associated to each gene and located in the promoter region (left), 10–100 kb upstream of gene TSS (middle) or at long genomic distances ( >100 kb) from the gene body. **f** Enriched transcription factor consensus matrix identified in HySp5 and ZfSp5a bound elements. **g** Principal component analysis showing the segregation of RNA-seq samples across the two main principal components. **h** Heat map plots showing the z score value of genes significantly up- or downregulated (based on Wald test $p < 0.05$) in HySp5 or ZfSp5a transfected HEK293T cells compared to their respective control conditions (HySp5-ΔDBD or ZfSp5a-ΔDBD). **i** Venn diagrams showing the number of HySp5 and ZfSp5a direct transcriptional targets (see description in **a**) significantly up- or downregulated

during evolution. This is surprising, since both HySp5 and mammalian orthologs can interact with β-catenin (Fig. 5f) to promote target gene activation. As the HEK293T cells are largely insensitive to huΔβ-Cat overexpression (Fig. 6g), the observed upregulation of HySp5 and ZfSp5a direct targets relies on mechanisms largely independent of β-catenin signaling. By contrast, the overexpression of HySp5 and ZfSp5a in zebrafish embryos leads to similar developmental alterations, which resemble those produced by the over-activation of Wnt/β-catenin signaling (Supplementary Fig. 18, Supplementary Data 4).

## Discussion

Studies performed in developing vertebrates show that *Sp5* is a target of Wnt/β-catenin signaling as recorded in zebrafish[27,28], mice[29], *Xenopus*[30], as well as in self-renewing mouse and human ESCs[31,37]. In line with these results, we show that in *Hydra*, *Sp5* is positively regulated by Wnt/β-catenin signaling as evidenced by its upregulation when Wnt/β-catenin signaling is pharmacologically enhanced. These results illustrate the deep conservation of the Wnt/β-catenin-dependent regulation of *Sp5* across eumetazoans. Wnt5, another candidate identified in the screen might also play a role in head inhibition, as a putative inhibitor of the canonical Wnt pathway[43,44] and a possible HySp5 target gene. By contrast, secreted Wnt antagonists such as Dickkopf (Dkk)[45] or Notum[46], both expressed in *Hydra*, were not identified in this screen.

*Wnt3* and *Sp5* upregulations in head-regenerating tips are consistent with a rapid head organizer formation after bisection. *Sp5* is re-expressed early during head regeneration, although as expected, later than *Wnt3*. This temporal parameter is indeed essential for the establishment of a de novo head organizer as demonstrated by transplantation experiments that accurately measured the successive re-activation of the two head organizer components, with head activation restored within 12 hpa and head inhibition coming back later, detectable at 24 hpa[9,13]. Here we used the qRNA-seq data to compare the respective regulations of *Wnt3* and *Sp5* in regenerating tips after decapitation or midgastric bisection. While *Wnt3* is rapidly upregulated to reach a plateau value at 4 hpa, *Sp5* shows an initial drop in expression within the first two hours following bisection, then an upregulation and a peak of expression detected at 8 hpa, four hours after that measured for *Wnt3*. If one assumes that the reestablishment of active Wnt3 and Sp5 proteins follows similar kinetics, then this four hour time window corresponds to a period when Wnt3/β-catenin signaling is active but Sp5 still inactive as *Wnt3* repressor, leaving sufficient time to instruct tissues to form a head.

A recent observation suggested that human SP5 can directly repress the *WNT3* promoter in human ESCs[37]. Here we demonstrate that indeed Sp5 from *Hydra* and zebrafish inhibit Wnt/β-catenin signaling by repressing the activity of the *Wnt3* promoter. Both the RNA-seq and the ChIP-seq data presented here confirm this view, by showing firstly that HySp5 and ZfSp5a when overexpressed in HEK293T cells repress largely overlapping sets of genes and secondly that both *Hydra* and zebrafish Sp5 preferentially bind genes of the Wnt/β-catenin signaling pathway, as observed in the promoter and intronic regions of the human *WNT3* and *WNT9B* genes. The studies performed in HEK293T cells also highlighted the fact that HySp5 and ZfSp5a, as transcriptional repressors, likely bind to regulatory elements located in the proximity of the TSS of their target genes. All together, these results highlight the similarity between the repressor effect of cnidarian and vertebrate Sp5 transcription factors, which predominantly affects genes of the Wnt/β-catenin signaling pathway but is not restricted to it. It is thus tempting to speculate that the Sp5-dependent inhibition of Wnt/β-catenin

signaling originated early in metazoan evolution and was maintained across eumetazoans. By contrast, the properties of HySp5 and ZfSp5a as transcriptional activators appear quite different: both can promote gene activation through β-catenin interaction, but they largely differ in their capacity to activate target genes in a β-catenin-independent mode. Therefore, we speculate that Sp5 possibly evolved the capacity to interact with partners not previously identified such as Tbx or Sox, and/or acquired the capacity to bind consensus motifs such as those enriched in the vertebrate long-range enhancers, after Cnidaria divergence.

Consistent with its *Wnt3* repressor function, *HySp5* silencing triggers in a highly robust way the ectopic formation of clusters of *Wnt3*-expressing cells, followed by the formation of multiple heads along the body column of intact animals, in head-regenerating regions and in reaggregates (Fig. 7). This phenotype is different from the ones obtained with pharmacological treatments, either with the GSK3-β inhibitor ALP[22,23,33] or recombinant Wnt3 that directly enhances β-catenin signaling[18,47], where ectopic tentacles form first, and heads appear several days later. In intact animals, the knockdown of *HySp5* leads to the direct and rapid formation of fully functional ectopic heads, preferentially in the budding zone, a region that is developmentally competent in adult animals where the expression of both *Wnt3* and *β-catenin* is quite dynamically regulated[17,18]. By increasing the number of dsRNA electroporations, we noted the formation of ectopic heads in the apical half of the body column, even though the development of these heads remained incomplete. Nevertheless, we never observed supernumerary heads at the apex of homeostatic *HySp5*(RNAi) animals, likely reflecting the difficulty to obtain a significant silencing in the apical region where *Sp5* expression is high. In the peduncle and basal part of the animal, ectopic head formation upon *HySp5*(RNAi) does not occur either, most likely as the physiological activity of Wnt3/β-catenin signaling is too low in this region to elicit ectopic head formation when *Sp5* is silenced. In head-regenerating animals or reaggregates, the *Sp5*(RNAi) phenotype is readily observed as, similarly to the budding zone, the expression of *Wnt3*, *β-catenin* and *Sp5* is quite dynamically regulated.

To further investigate these dynamic modulations, we designed strategies to modulate the *Sp5*(RNAi) phenotype. We first noticed that when *β-catenin* is silenced, the *Sp5*(RNAi) phenotype is greatly reduced, indicating that an active Wnt3/β-catenin signaling is necessarily required for ectopic head formation. We also measured the spatial spreading of the ALP-induced phenotype when *Sp5* is knocked-down, with ectopic *Wnt3* expression and ectopic tentacle formation all along the body column. This last result indicates that the constitutive activation of Wnt3/β-catenin signaling by ALP is significantly enhanced upon *Sp5* silencing. These modulations of the *Sp5*(RNAi) phenotype in response to *β-catenin*(RNAi) or the ALP-induced phenotype in response to *Sp5* (RNAi) again confirm the intimate dynamic cross-talk that takes place between *Sp5* regulation, Sp5 activity and the Wnt3/β-catenin signaling activity.

The observed *Sp5*(RNAi) phenotypic modulations indicate that *Sp5* silencing cannot be easily maintained stable along the midgastric region, namely because its regulation is quite dynamic in response to the level of Wnt3/β-catenin signaling. Therefore, we interpret the homeostatic *HySp5*(RNAi) phenotype in the budding region as the consequence of the transient downregulation of *HySp5* activity in tissues that have the highest potential for setting up an organizer as evidenced by the transient upregulation of β-catenin in the budding zone[17]. As an evidence of this dynamic cross-talk, we noticed that a transient drop in *HySp5* expression suffices to rapidly induce a de-repression of *Wnt3* expression, which leads to an upregulation of β-catenin activity, and in turn

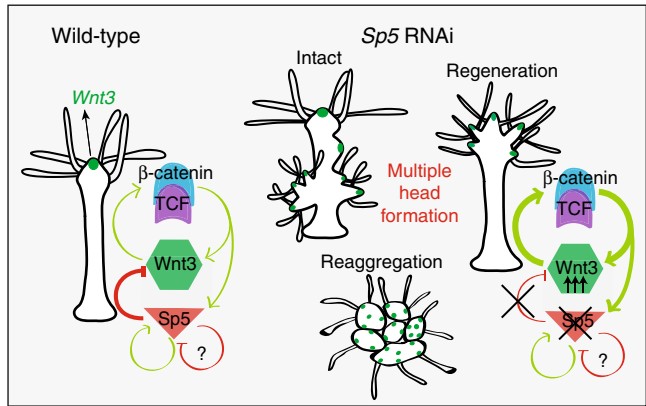

**Fig. 7** Working model of the feedback loop involving Wnt3/β-catenin/TCF and Sp5. Wnt/β-catenin signaling positively regulates *Wnt3* expression via β-catenin stabilization as well as *Sp5* expression. Head organizer activity is restricted by Sp5 that positively auto-regulates its own expression, likely by interacting with the β-catenin/TCF complex, but also represses the expression of *Wnt3* through the *Wnt3* repressor element. Depending on the level of Sp5 in a cell, Sp5 might also repress its own expression. This tight transcriptional control mechanism might then ensure a stable repression of the *Wnt3* promoter. In the absence of Sp5, the repressing effect on the *Wnt3* promoter is lost and *Wnt3* is no longer restricted to the head organizer. The release of *Wnt3* expression is sufficient to trigger multiple head formation in intact and regenerating conditions as well as in reaggregates

to *Wnt3* upregulation followed by that of *Sp5* (Fig. 7). The oscillatory nature of *HySp5* and *β-catenin* expression in regions competent for head organizer formation suggests a bistability state relying on an auto-regulatory loop involving two transcription factors[48]. This bistability as a prerequisite to head organizer induction and/or head organizer maintenance remains to be explored.

This study identifies the transcription factor Sp5 as a key inhibitory component of the *Hydra* head organizer. Indeed Sp5 fulfills the five criteria we initially fixed, derived from the predicted properties of the head inhibitor and from the previous identification of Wnt/β-catenin signaling as the head activator[19]. Sp5 globally fits the Turing/Gierer-Meinhardt model as *HySp5* expression is controlled by Wnt3/β-catenin signaling, predominantly expressed in the head, reactivated during head regeneration, while HySp5, as a *Wnt3* repressor, represses ectopic head formation (Fig. 7). However, several features diverge from the expected properties of the head inhibitor predicted by the Gierer-Meinhardt model.

Firstly, we noted the lack of *Sp5* expression at the very apical tip of the hypostome in intact animals, the region where *Wnt3* expression, and most likely Wnt3 activity, is maximal. Two distinct cis-regulatory elements in the *Wnt3* promoter were previously identified, an activator and a repressor element, the latter restricting *Wnt3* expression to the distal tip of the head[19]. The *Sp5* pattern is thus consistent with the prediction that the inhibitor should be absent or unable to repress *Wnt3* in this area. As *Sp5* appears as a direct target of Wnt3/β-catenin signaling (see below), an additional negative regulation has to take place in this most apical area, to prevent *Sp5* expression. This local regulation remains to be identified.

Secondly, this study supports a scenario where Wnt3 acts as a short-range activator to sustain its own activity in the head organizer, while Sp5 prevents the expression of *Wnt3* and possibly other *Wnt* genes in non-apical tissues. The Gierer-Meinhardt model, proposed at a time when the concept of transcription factor

was still unknown, predicts that the head inhibitor is a diffusible substance, acting non-cell autonomously across the tissue layers. As a transcription factor, HySp5 is suspected to act cell-autonomously and thus not diffusible. However, some transcription factors can be secreted, as reported for the helix-turn-helix transcription factor EspR in bacteria[49] or for some homeoproteins that exert non-cell autonomous functions in the mammalian brain[50]. Also, Sp5 might upregulate target genes that encode secreted peptides or proteins that diffuse in the extra-cellular space and exert head inhibitory functions. Such target genes, possibly taxon-specific, remain to be identified.

Thirdly, we cannot exclude that Wnt signals, which are numerous to be emitted from the apical region[18] are not short-range signals but rather act over long-range distances to activate *HySp5* expression with lipid-binding proteins or cytonemes modulating the spread of Wnt proteins as observed in *Drosophila*, *Xenopus* and zebrafish[51–53]. The inhibition of Wnt3/β-catenin signaling along the *Hydra* body axis might thus solely be mediated by transcriptional repression, with Sp5 regulating its own expression and tightly tuning the level of Wnt signals.

As a fourth divergence with the Gierer-Meinhardt model, we found that HySp5 activates its own promoter. Both the reporter assays and the ChIP-qPCR data demonstrate that HySp5 directly binds its own promoter, while the ChIP-seq data also suggest that HySp5 is able to bind the human SP5 promoter. These observations are consistent with a study showing that the mouse Sp5 protein directly binds and activates its own promoter[31]. In addition, β-catenin slightly enhances the activating effect of HySp5 on its promoter, likely through direct interaction between HySp5, TCF1 and/or β-catenin as observed in vitro. A recent study demonstrates a direct interaction between the zinc finger domain of mouse Sp5 and the HMG domain of Tcf/Lef1, while no direct interaction was observed for β-catenin[31]. Also the formation of active Tcf/Lef1-β-catenin complexes appears necessary for Sp5 DNA-binding in mouse ESCs[31]. In contrary, in human ESCs, SP5 could directly repress the human *SP5* promoter[37]. Thus, currently we cannot exclude that besides its auto-activating effect, HySp5 might also have an auto-repressing effect when it reaches high intracellular levels for example. Further studies should evidence this putative auto-repressing effect as well as the interactions between HySp5 and TCF/β-catenin that favor the switch from Sp5 auto-activation to Sp5 auto-repression.

## Methods

**Animal culture and drug treatment**. All experiments were carried out with *Hydra vulgaris (Hv)* from the Basel, AEP or Hm-105 strains. Cultures were maintained in Hydra Medium (HM: 1 mM NaCl, 1 mM CaCl₂, 0.1 mM KCl, 0.1 mM MgSO₄, 1 mM Tris pH 7.6) or in Volvic water, supplemented with 0.5 mM CaCl₂. Animals were fed two to three times per week with freshly hatched *Artemia* nauplii and starved for four days before any experiment. For drug treatments Hv_Basel were treated for two days with 5 μM Alsterpaullone (ALP, Sigma) diluted in HM, 0.015% DMSO then rinsed 3x in fresh HM. All animals were selected randomly for experiments.

**Generation of the *HyWnt3*:GFP-*HyAct*:dsRED transgenic strain**. To induce gametogenesis, *H. vulgaris* of the strain AEP were fed with freshly hatched *Artemia* nauplii 7× per week for three weeks and then 1× per week for 1 week. Thereafter, male and female animals were cultured together, resulting in fertilized embryos. The hoTG-HyWnt3FL-EGFP-*HyAct*:dsRED plasmid (kind gift from T. Holstein, Heidelberg)[19] was injected into one-cell stage embryos. Out of 504 injected eggs, 104 embryos hatched and 7/104 embryos exhibited GFP fluorescence in the hypostome.

**RNA interference**. In short, intact *Hydra* were briefly washed and incubated for 45 min in Milli-Q water[54]. 20 animals per condition were placed in 200 μl 10 mM sterilized HEPES solution (pH 7.0) and then transferred into a 0.4 cm gap electroporation cuvette (Cell Projects Ltd). Animals were electroporated with 4 μM of Sp5 (siRNA-1+siRNA-2+siRNA-3) or scramble siRNAs (Supplementary Table 1b) using the Biorad GenePulser Xcell electroporation system. For double knockdown experiments 2 μM of Sp5 siRNAs were mixed with 2 μM of scramble of

β-catenin siRNAs. The conditions of electroporation were: Voltage: 150 Volts; Pulse length: 50 milliseconds; Number of pulses: 2; Pulse intervals: 0.1 s. For subsequent ALP treatment, RNAi animals that did not show any phenotypic signs were kept for 18 h in HM containing 5 μM ALP. The animals were then relaxed in 2% urethane/HM for one minute, fixed in 4% PFA prepared in HM (pH 7.5) for 2 h at RT and either processed for WISH, or directly mounted with Mowiol for picturing.

**Reaggregation**. Animals were electroporated twice (RNAi1, RNAi2) with siRNAs and treated with ALP as described above. Next, 50–60 animals of the same size that did not show any phenotypic signs, were dissociated in 10 mL of dissociation medium (DM) (3.6 mM KCl, 6 mM CaCl₂, 1.2 mM MgSO₄, 6 mM Na-Citrate, 6 mM Pyruvate, 4 mM Glucose and 12.5 mM TES; pH 6.9)[55] and the cell suspension was centrifuged at 1'400 rpm for 30 min at 4 °C. The pellet was resuspended in 1 mL DM and 450 μl of the cell suspension equally distributed into 1.5 mL tubes, followed by centrifugation at 1'400 r.p.m. for 30 min. After detachment, the aggregates were kept for one hour at 18 °C in 75% DM/HM and overnight in 50% DM/HM. On the next day, the aggregates were transferred into HM.

**Quantitative RT-PCR**. Total RNA was extracted using the E.Z.N.A.® Total RNA kit (Omega) and cDNA synthesized using the qScript™ cDNA SuperMix (Quanta Biosciences). qPCR was performed in a 96-well format using the SYBR™ Select Master Mix for CFX (Thermo Fisher Scientific) and a Biorad CFX96™ Real-Time System. The *TBP* gene was used as an internal reference gene. HySp5 and TBP primer sequences are listed in Supplementary Table 1c. For ChIP-qPCR, DNA was prepared as below (ChIP-seq section) and qPCR performed as above with primer sequences listed in Supplementary Table 1d.

**Whole mount In Situ Hybridization and immunodetection**. *Hydra* were relaxed in 2% urethane/HM for one minute, fixed in 4% PFA prepared in HM (pH 7.5) for 4 h at RT and stored in MeOH at −20 °C for at least one day. Samples were rehydrated through a series of EtOH, PBSTw (PBS, Tween 0.1%) washes (75%, 50%, 25%) for 5 min each, washed 3× with PBSTw for 5 min, digested with 10 μg/mL Proteinase K (PK, Roche) in 0.1% SDS, PBSTw for 10 min, stopped by adding Glycine (4 mg/mL) and incubated for 10 min. Samples were washed 2x in PBSTw for 5 min, treated with 0.1 M TEA for 2 × 5 min, incubated 5 min after adding acetic anhydride 0.25% (v/v), 5 min after adding again acetic anhydride 0.25% (final concentration 0.5% v/v). Samples were then washed in PBSTw 2 × 5 min, post-fixed in 4% formaldehyde, PBSTw for 20 min, washed in PBSTw 4 × 5 min before adding the pre-warmed pre-hybridization buffer (PreHyb: 50% Formamide, 0.1 % CHAPS, 1× Denhardt's, 0.1 mg/mL Heparin, 0.1% Tween, 5x SSC) and incubated for 2 h at 58 °C. Next, 350 μL hybridization buffer (PreHyb containing 0.2 mg/mL t-RNA, 5% Dextran) containing 200 ng DIG-labeled riboprobe was heated 5 min at 80 °C, then placed on ice for 2 min. This mix was added onto the samples, then incubated for 19 h at 58 °C. Next, the samples were rinsed 3x in pre-warmed PostHyb-1 (50% formamide, 5x SSC) and successively incubated for 10 min at 58 °C in PostHyb-1, PostHyb-2 (75% PostHyb-1, 25% 2x SSC, 0.1% Tween), PostHyb-3 (50% PostHyb-1, 50% 2x SSC, 0.1% Tween) and PostHyb-4 (25% PostHyb-1, 75% 2× SSC, 0.1% Tween). Samples were then washed 2 × 30 min in 2× SSC, 0.1% Tween, 2 × 30 min in 0.2x SSC, 0.1% Tween, 2 × 10 min in MAB-Buffer1 (1× MAB, 0.1% Tween), blocked in MAB-Buffer2 (20% sheep serum, MAB-Buffer1) for 1 h and incubated with anti-DIG-AP antibody (1:4000, Roche) in MAB-Buffer2 overnight at 4 °C. Next, the samples were washed in MAB-Buffer1 for 4 × 15 min, then in NTMT (NaCl 0.1 M, Tris-HCl pH 9.5 0.1 M, Tween 0.1%) for 5 min and finally in NTMT, levamisole 1 mM for 2 × 5 min. The colorimetric reaction was started by adding staining solution (Tris-HCl pH 9.5 0.1 mM, NaCl 0.1 mM, PVA 7.8%, levamisole 1 mM) containing NBT/BCIP (Roche). The background color was removed by a series of washes in EtOH/PBSTw (30%/70%, 50%/50%, 70%/30%, 100% EtOH, 70%/30%, 50%/50%, 30%/70%), PBSTw 2 × 10 min. Samples were post-fixed for 20 min in FA 3.7% diluted in PBSTw, washed in PBSTw 3 × 10 min and mounted with Mowiol. All steps were performed at RT unless indicated otherwise. Whole mount immunofluorescence with the anti-RFamide antibody (kind gift of C. Grimmelikhuijzen, 1:1000) was performed as in ref. [32].

**Peroxidase assay**. *Hydra* were relaxed in 2% urethane/HM for one minute and fixed in 4% PFA prepared in HM (pH = 7.5) for 2 h at RT. Samples were washed 3 × 10 min with PBS, followed by adding 500 μL DAB (SIGMAFAST™ 3,3'-Diamino-benzide) solution. The DAB solution was prepared as follows: 1 tablet of DAB was dissolved in 10 mL of PBS and filtered with a 0.22 μm filter. 5 mL of the filtered solution was added to 5 mL of PBS together with 20 μL of Triton X-100 (0.2%) and 1 μL of a 30% H₂O₂ solution. The animals were incubated for 10 min in DAB solution and the reaction stopped by washing the samples 3 × 10 min with PBS.

**Plasmid constructions**. To generate the *HyWnt3*:Luc construct 2149 bp of the *Hydra Wnt3* promoter were transferred from the hoTG-HyWnt3FL-EGFP construct (kind gift from T. Holstein, Heidelberg)[19] into the pGL3 reporter construct (kind gift from Z. Kozmik, Prague)[29]. For the *HyWnt3*-ΔRep:Luc construct, the whole *HyWnt3*:Luc plasmid sequence was PCR-amplified except the 386 bp

corresponding to the repressor element. For the *ZfWnt3*:Luc construct 3997 bp of the zebrafish *Wnt3* promoter were transferred from pEGFP-Wnt3 (kind gift of Cathleen Teh, Singapore) into pGL3. For the *HySp5*:Luc construct, 2'992 bp of the *Hydra Sp5* promoter were PCR-amplified from *Hm-105* genomic DNA and subcloned into pGL3. To express HA-tagged HySp5, ZfSp5a, ZfSp5l1 proteins, a C-terminal HA-tag was introduced into the pCS2 + constructs encoding the *Hydra Sp5* (human codon-optimized), zebrafish *Sp5a* and *Sp5l1* full-length coding sequences. The HySp5-ΔSP construct was produced by inserting a human codon-optimized *HySp5* sequence lacking 110 amino acids of the N-terminal end together with a C-terminal HA-tag into pCS2+. The HySp5-ΔDBD and HySp5-ΔSP-ΔDBD constructs were generated using the QuikChange Lightning Multi Site-Directed Mutagenesis Kit (Agilent Technologies), following the manufacturer's instructions. To generate the ZfSp5a-ΔDBD construct, the ZfSp5a-FL plasmid sequence was PCR-amplified except the DNA-binding domain. For preparing riboprobes, the *HyWnt3, HyBra1, HyTsp1, HyKazal1* and *HySp5* PCR products were cloned into pGEM-T-Easy (Promega). All constructs were verified by sequencing. All plasmids are listed in Supplementary Table 2 and primer sequences in Supplementary Table 1a.

**Reporter assays in human HEK293T cells**. HEK293T cells were maintained in DMEM High Glucose, 1 mM Na pyruvate, 6 mM L-glutamine, 10% fetal bovine serum. For the luciferase assays HEK293T cells were seeded into 96-well plates (5000 cells/well) and transfected 18 h later with X-tremeGENE™ HP DNA transfection reagent (Roche). The plasmids listed in Supplementary Table 2 were transfected as follows: pGL4.74(hRluc/TK) (Promega): 1 ng, luciferase reporter constructs: 40 ng, *CMV*:huΔβ-Cat: 10 ng, Sp5 expression constructs: 20 ng, huWnt3 and huLRP6: 40 ng. Total DNA amount was adjusted with pTZ18R to 100 ng per well. To measure Firefly and *Renilla* luciferase activities, the samples were prepared using the Dual-Luciferase Reporter Assay System (Promega), transferred to a white OptiPlate™-96 (PerkinElmer) and measured with a multilabel detection platform (CHAMELEON™).

**ChIP-seq sample preparation**. 920'000 HEK293T (92 cells/μL) cells were seeded into a 10 cm dish containing 10 mL of cell culture medium and transfected as described above with HySp5 or ZfSp5a, both containing a C-teminal HA tag (3'666 ng). Twenty-four hours later, cells were collected, washed twice in pre-warmed culture medium, fixed in 1% formaldehyde (FA) solution (Sigma) for 15 min until Glycine was added (final 125 mM) for 3 more minutes. In subsequent steps numerous reagents were from Active Motif™ (AM). The cells were washed once in ice-cold PBS and re-suspended in 5 mL chromatin prep buffer (AM), containing 0.1 mM PMSF and 0.1% protease inhibitor cocktail (PIC). The sample was transferred into a pre-cooled 15 mL glas Douncer, dounced with 30 strokes and incubated on ice for 10 min. Nuclei were centrifuged at 1250 g for 5 min at 4 °C, resuspended in 500 μL sonication buffer (1% SDS, 50 mM Tris-HCl pH 8.0, 10 mM EDTA pH 8.0, 1 mM PMSF, 1% PIC), incubated on ice for 10 min. Next, the chromatin was sonicated with a Bioblock Scientific VibraCell 75042 sonicator (Amplitude: 25%, Time: 12 min, 30 s on, 30 s off, 24 cycles), in conditions optimized to have a fragmentation size of ~250 bp. Then 100 μL of the sonicated chromatin was added to 900 μL ChIP dilution buffer (0.1% NP-40, 0.02 M HEPES pH 7.3, 1 mM EDTA pH 8.0, 0.15 M NaCl, 1 mM PMSF, 1% PIC) and incubated with 4 μg anti-HA antibody overnight at 4 °C on a rotator. Next, the sample was loaded on a ChIP-IT ProteinG Agarose Column (AM), incubated for 3 h at 4 °C on a rotator, washed 6x with 1 mL AM1 buffer and the DNA eluted with 180 μL pre-warmed AM4 buffer. The sample was decrosslinked by adding 100 μL high salt buffer (1 M NaCl, 3× TE buffer) and incubated for 5 h at 65 °C. RNAse A (10 μg/μL) was added and the sample incubated at 37 °C for 30 min before adding PK (10 μg/μL) and further incubated for 2 h at 55 °C. The DNA was purified with the MiniElute PCR purification kit (Qiagen). For preparing the Input DNA, 5 μL sonicated chromatin was diluted in 45 μL 0.5 M NaCl, incubated for 15 min at 95 °C, then transferred to 37 °C, incubated for 5 min with RNAse A (10 μg/μL), adding PK (10 μg/μL) and incubated at 55 °C for 30 min. 10 μL were taken for purification (MiniElute PCR purification kit from Qiagen).

**RNA-seq sample preparation**. 156'500 HEK293T (78.25 cells/μL) cells were seeded into a 6-well plate containing 2 mL of cell culture medium and transfected as described above with 626 ng of HySp5, ZfSp5a, HySp5-ΔDBD, ZfSp5a-ΔDBD and 313 ng of human Δβ-Catenin. RNA was extracted with the E.Z.N.A. total RNA kit I from OMEGA following the manufacturer's instructions.

**Co-immunoprecipitation assay and Western blotting**. 920'000 HEK293T cells (92 cells/μL) were seeded into a 10 cm dish containing 10 mL of cell culture medium and transfected with huΔβ-Cat (1830 ng), huTCF1 (1830 ng) and HySp5 (3660 ng). 24 h later, Co-IP samples were prepared using the nuclear complex Co-IP kit from Active Motif, following the manufacturer's instructions (all steps at 4 °C with ice-cold buffers). 100 μg nuclear extracts were then diluted in 500 μL Co-IP incubation buffer containing 4 μg anti-HA antibody or 4 μg rabbit IgG (12–370, Merck Millipore) and incubated overnight on a rotator. The Co-IP reaction was then loaded on a Protein G Agarose column (AM) and incubated one hour on a rotating wheel. The column was washed 3x in 500 μL Co-IP wash buffer

supplemented with 1 mg/mL BSA, 3x in 500 μL of Co-IP wash buffer supplemented with 300 mM NaCl. The column was centrifuged at $1250 \times g$ for 3 min and 25 μL 2x reducing buffer directly added onto the column. After 60 s incubation and 3 min centrifugation at $1250 \times g$, 5 μL glycerol (Sigma) was added and the sample boiled for 5 min at 95 °C before loading on a 8% SDS-PAGE gel, electrophoresed and transferred onto PVDF membrane (Bio-Rad). Then all steps were performed at RT unless specified. The membrane was blocked with M-TBS-Tw (TBS containing 0.1% Tween, 0.5% dry milk) for one hour until primary antibodies diluted 1:2000 in M-TBS-Tw were added for overnight incubation at 4 °C. The membrane was then washed $4 \times 10$ min in TBS-Tw, incubated in anti-rabbit (ab99697, Abcam) or anti-mouse (W402B, Promega) IgG horseraddish peroxidase antibody (1:5000) for one hour, visualized with Western Lightning® Plus-ECL reagent (PerkinElmer). 10 μg extract were used as Input sample. Antibodies: anti-HA antibody (NB600–363, Novus Biologicals), anti-β-catenin antibody (610153, BD Biosciences), anti-TCF1 (sc-271453, Santa Cruz Biotechnology). All uncropped western blots can be found in Supplementary Fig. 15.

**ChIP-seq data analysis.** Demultiplexed ChIP-seq reads from our sequenced samples were mapped onto the Human GRCh37 (hg19) genome assembly using bowtie2, version 2.2.6.2[56], implemented in galaxy[57]. Significantly enriched regions were identified using MACS2[58] (version 2.1.0.20151222.0). Coverage files were normalized by the millions of mapped reads in each sample using a manually created R script. Normalized bedgraph files were converted to bigwig using the Wig/BedGraph-to-bigWig converter tool (version 1.1.1) implemented in the pubblic Galaxy server (https://usegalaxy.org/) and visualized with UCSC genome browser. The fastq files from the two biological replicates of each condition were merged and remapped in order to obtain the average coverage profile. Only autosomal chromosomes were analysed in this study. MACpeaks regions were either extended or cropped from their respective center to match a final size of 500 bp using a personalized R script based on the GenomicRanges package (version 1.32.6). Fasta files containing the DNA sequences corresponding to the coordinates of the MACpeaks regions were obtained using the UCSC table browser tool. These files were used to identify enriched motifs for transcription factor binding sites using the MEME-ChIP Suite[59] (http://meme-suite.org/tools/meme-chip) in classic mode. Significantly enriched motifs were identified and compared to previously described TF weight matrixes from the JASPAR CORE 2014 database[59] using the TOMTOM tool of the MEME-ChIP suite. Significantly enriched motifs were used to scan the HySp5, HyWnt3 and ZfWnt3 promoters, using the FIMO tool (http://meme-suite.org/tools/fimo)[60] to identify putative Sp5 binding sites. Gene assignment of the identified MACpeak region was performed using the ChipEnrich Package in R (version 2.4.0; locus definition: nearest TSS; gene set: gene ontology biological process; method: polyenrich). Calculations of the total HySp5 and ZfSp5a coverages (in Mb) and of the frequency distribution of the number of Sp5-enriched regions per gene were performed in R using personalized scripts. ChIP-seq data sets for the Sp5 and β-catenin occupancies in mouse ES cells[31,61] were, respectively, downloaded from the GEO subseries GSE72989 and GSM1065517 and re-mapped on the mouse mm10 genome assembly using the same workflow describe above.

**RNA-seq data analysis.** Demultiplexed RNA-seq reads from our sequenced samples were mapped onto the Human GRCh37 (hg19) genome assembly using the STAR RNA-seq aligner[62] workflow implemented in Galaxy. The fastq files from the three biological replicates of each condition were merged and remapped in order to obtain the average coverage profile. Coverage files were normalized by the millions of mapped reads in each sample using a manually created R script. Normalized bedgraph files were converted to bigwig using the Wig/BedGraph-to-bigWig converter tool implemented in the public Galaxy server (https://usegalaxy.org/) and visualized with UCSC genome browser. We used Htseq[63] implemented in the Galaxy server to count the number of uniquely mapped reads attributable to each gene (based on human genomic annotations from Ensembl release 82[64]). We used DESeq2[65] to perform differential expression analyses. Specifically, we contrasted a generalized linear model that explains the variation in read counts for each gene, as a function of the different transfection conditions, to a null model that assumes no effect of the HySp5/HySp5-ΔDBD, ZfSP5a or ZfSp5a-ΔDBD. We ran the Wald test and the $P$ values were corrected for multiple testing with the Benjamini–Hochberg approach. We computed reads per kilobase of exon per million mapped reads gene expression levels using Cufflinks[66].

FPKM levels were Log2-transformed, after adding an offset of 1 to each value. The Log2-transformed values were centered across samples before Principal Component Analysis (PCA); no variance scaling was performed. Heatmap plots were produced using the gplot package (version 3.0.1 in R). For this we computed the Z score $((X - \mu)/\sigma$, where for each gene $\mu$ and $\sigma$ are respectively the average and standard deviation of all the replicates of the two conditions being compared and $X$ is the FPKM value of each sample) based on the FPKM value of each gene differentially expressed between HySp5 vs HySp5-ΔDBD or between ZfSp5a vs ZfSp5a-ΔDBD. Up- and downregulated genes from this analysis were considered as HySp5 and/or ZfSp5a putative targets if they were associated with a MACpeak enriched region for these proteins (based on the chipenrich analysis described above).

**GO term enrichment analysis.** We used the GOrilla tool[67] to search for enriched GO term categories associated with HySp5/ZfSp5a bound genes and with upregulated or downregulated HySp5/ZfSp5a putative targets using a treshold of $p < 10^{-3}$ (FDR < 0,05). In the latter case, when more than 10 significantly enriched GO term categories were identified, we used the REVIGO tool[68] using 0,7 as treshold for allowed similarity between related GO term classes.

***Hydra* genome assembly.** Five clonal animals of the species *Hydra viridissima* and *Hydra oligactis* were sampled independently to extract DNA material using the DNeasy Blood & Tissue kit (Qiagen). Sequencing libraries were prepared using the TruSeq Nano DNA kit (Illumina), with 350 bp insert sizes, and sequenced paired-end using 150 cycles on an Illumina HiSeq X Ten sequencer by Macrogen Inc. Average and standard deviations of insert sizes of the sequenced reads were measured using 10 mio reads mapped to a preliminary assembly of each genome, then the two genomes were assembled using MaSuRCA v3.2.1[69]. All scaffolds (>300 bp) and unplaced contigs (>500 bp) were retained in the final set of sequences. The redundancy of each assembly was reduced by using CD-HIT-est v4.7[70] with a 100% identity threshold. Sequencing depth was evaluated from the number of reads and expected genome length: *Hydra viridissima*: 120×; *Hydra oligactis*: 50×. Scaffolds assembly statistics in bp: number of scaffolds: 85677 for *viridissima* and 447337 for *oligactis*; N50: 11871 for *viridissima* and 5391 for *oligactis*.

***Hydra* RNA-seq transcriptomics.** For spatial and cell-type RNA-seq transcriptomics, see ref. [32]. All profiles publicly available on the HydrATLAS server (https://HydrATLAS.unige.ch).

**Multiple sequence alignment and phylogenetic analysis.** For Supplementary Figure 2, the multiple sequence alignment was generated using T-Coffee[71]. The conserved zinc finger domains, SP and Btd boxes were visualized by IBS[72]. For the phylogenetic analysis of the Sp5, Sp-related and Klf-related gene families (Supplementary Figure 3), sequences from *Hydra* as well as from other cnidarian, ecdysozoans, lophotrochozoans and deuterostomes representative species were retrieved from Uniprot or NCBI, aligned with Muscle align (www.ebi.ac.uk/Tools/msa/muscle/)[73] or MAFFT (https://mafft.cbrc.jp/alignment/server/) and tested in iterative PhyML 3.0 analyses using the LG substitution model, 8 substitution rate categories and 100 bootstraps[74].

**Sp5 expression in zebrafish embryos.** For all zebrafish experiments, colonies of the strain AB-Tu or Nacre were used, with animals maintained at 28 °C with a maximal density of five fish per liter in a 14 h light–10 h dark cycle. The fish were fed twice a day with 2-day-old *Artemia* and fish embryos incubated at 28 °C. For overexpression experiments, capped sense mRNAs were synthesized using the mMESSAGE mMACHINE® Transcription Kit from Ambion (Ambion, Austin, TX USA) and 400 pg of *HySp5, HySp5-ΔDBD, HySp5-ΔSP* or *HySp5-ΔSP-ΔDBD* mRNAs injected into one cell stage embryos. For mRNA co-injection experiments, injected amounts were as follows: 400 pg of HySp5 and 4 pg of ZfWnt8 mRNA. All embryos were scored for phenotypes 48 h post fertilization.

**Statistical analyses.** All statistical analyses were performed with the software GraphPad Prism7. The statistical tests were two-tailed unpaired.

**Reporting Summary.** Further information on experimental design is available in the Nature Research Reporting Summary linked to this article.

## Data availability

The *Hydra Sp5* sequence has been deposited in GenBank under: MG437301. The genome assemblies and reads have been deposited in the BioProject under: PRJNA419866. RNA-seq and ChIP-seq data have been deposited in the GEO database under accession code GSE121321 [https://www.ncbi.nlm.nih.gov/geo/query/acc.cgi?acc = GSE121321]. The authors declare that all data supporting the findings of this study are available within the article and its supplementary information files or from the corresponding author upon reasonable request. The Source Data underlying Figs. 4f, 5d and Supplementary Figs. 6b, 12d are provided as a Source Data file.

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

## Acknowledgements

This work was supported by the Swiss National Foundation (SNF 31003A_149630; 31003_169930), the Canton of Geneva, the Human Frontier Science Program (grant no. RGP0016/2010), the NCCR 'Frontiers in Genetics', the Claraz donation and the de Staël foundation. The authors warmly thank Denis Duboule and Claude Desplan for valuable inputs on the manuscript, Ariel Ruiz I Altaba and Charisios Tsiairis for discussions and reagents, Carol Gauron for excellent technical assistance, Nenad Suknovic and Szymon Tomczyk for help with image acquisition and the IGe3 Genomic Platform for ChIP-seq and RNA-seq library preparation and sequencing.

## Author contributions

M.C.V. performed *Hydra* and cell culture experiments, performed biochemical assays and prepared ChIP-seq and RNA-seq samples; L.B. analyzed ChIP-seq and RNA-seq data. M.C.V. and L.B. performed ChIP-qPCRs. L.I.O. contributed to plasmid constructions, knockdown experiments and in situ hybridizations; C.R. and S.V. performed zebrafish experiments; Y.W. and B.G. designed the *Hydra* high-throughput transcriptomics, Y.W. produced and processed the *Hydra* high-throughput transcriptomics as well as the genome data; C.P. produced the transgenic line; M.C.V. and B.G. conceived the study, M.C.V., L.B. and B.G. wrote the manuscript.

## Additional information

**Competing interests:** The authors declare no competing interests.

