## [Peer Review File · Nature Communications]

Reviewers' Comments:

Reviewer #1:

Remarks to the Author:

In this manuscript, the authors provide evidence for Sp5 being a head inhibitor in Hydra. This work confirms prior work in other organisms (zebrafish, mice and humans) that Sp5 is a Wnt target gene that is activated robustly upon Wnt pathway activation. Sp5 in turn acts to repress other critical Wnt target genes, including Wnt3, thereby serving as a negative feedback regulator of Wnt. The authors also provide evidence that Sp5 activates its own expression, a finding consistent with previously published work in mouse ES cells. Overall, this is an interesting study that addresses an important question of how the Hydra head organizer is regulated by interactions between an activator, Wnt3, and an inhibitor, Sp5.

As detailed below, this study has several issues that need to be addressed. More importantly, this study fails to put to rest a discrepancy in the analysis of Sp5 acting downstream of Wnt/b-catenin signaling: is Sp5 primarily a transcriptional repressor, a transcriptional activator, or both? The authors provide evidence that hySp5 can act as both, a repressor of such genes as Wnt3 and an activator of itself. How Sp5 achieves this dual function is unclear, and the literature provides conflicting results: Fujimora et al. (2007, PMID: 17090534) and Huggins et al. (2017, PMID: 29044119) provide evidence that Sp5 acts primarily as a repressor. In contrast, in mice, Kennedy et al. (2016, PMID: 26969725) argued that Sp5 acts to activate Wnt/b-catenin target gene expression. This discrepancy can certainly be rationalized many different ways. However, it would be preferred if the authors of this paper could address this question head on and possibly resolve at least Sp5's ancestral/original role as a repressor or activator or both. If it is in fact the case that Sp5 acts as both repressor and activator, it would be important to address how this happens. Interaction between Sp5 and b-catenin/Tcf has been shown by others and this could lead to either inhibition or activation. However, this fails to explain how Sp5 could achieve both functions as a repressor and activator. Something is missing and it would be good if the authors could shed some light on this.

Consistent with Sp5 being a repressor, the authors show that Sp5 knockdown (KD) by siRNA leads to ectopic Wnt3 expression. Here, the authors should test whether Sp5 expression is elevated or diminished upon Sp5 KD. Knockout of SP5 in human ES cells led to elevated expression of SP5, indicating that SP5 served to repress its own expression. A similar experiment in Hydra with siRNA KD may be difficult to examine by in situ hybridization for Sp5. Instead, the authors could test this with a transgenic Hydra carrying a Sp5 reporter, similar to the experiments shown in Fig. 2f and g with the HyWnt3:GFP animals. This experiment would require using the HySp5-2992 reporter (e.g. GFP) in Hydra and showing that this reporter is either activated (indicating that Sp5 is a repressor of itself) or inhibited (indicating that Sp5 is an activator of itself). Currently, the authors only provide evidence for Sp5 being an activator in a HEK293 reporter assay with overexpression of tagged HySp5.

To follow is a list of additional comments/critiques, not in any particular order, that need to be addressed:

Figure 3a-c: The ~300 fold increase in basal expression levels of the HyWnt3-1763 reporter compared to the HyWnt3-2149 reporter is surprising and unexplained. Do the authors think that this increase in expression is entirely due to loss of Sp5 binding sites, resulting in de-repression? This possibility should be explored further. The more refined deletions of this region indicate that the repressive elements reside in two ~100bp regions, one at -386/-286 and one at -95/-1. Extended Figure 7a indicates two predicted Sp5 binding sites reside in these regions (unfortunately, as illustrated, it is not possible to correlate the map shown in Ext.Fig.7a with the promoter elements shown in Figures 3a and c; this should be fixed); does mutation of these pBSs lead to loss of repression of Δ Rep-D3?

On line 117, the authors state "To test whether HySp5 autoactivates its own promoter, we preformed ChIP-qPCR experiments..." Such an experiment does not test whether Sp5 autoactivates; it shows that Sp5 can bind in the promoter of Sp5; whether Sp5 activates or inhibits is a separate question that is addressed in Fig. 3h.

The interpretation of Fig. 3h is a bit flawed: the authors claim that (line 122) "HySp5....increases activity of the HySp5 promoter, an effect enhanced upon b-catenin overexpression." This claim is not supported by the data: the level of trans-activation between HySp5 alone and HySp5+b-catenin are essentially the same, i.e. there is no enhancement of activity in the presence of b-catenin. It is also a little misleading and confusing that the authors split the data into 2 graphs: these should be combined into one single graph.

Figure 3i is unclear: what does inclusion of HySp5-2992:Luc (indicated by "+") mean? Does this co-IP require the presence of this reporter element? Will the interactions between Sp5 and b-catenin/Tcf1 be observed if this reporter element were not present? I assume so, but it is unclear why the figure is labeled as is. Also in this figure, it is unclear whether Sp5 co-immunoprecipitates with endogenous b-catenin or with the slightly truncated transfected version of b-catenin

Figure 1e: what do the numbers in the third column (under "Apical, R1, R2, R3, R4, Basal") signify? Are these RPKMs?

Figure 2f could be moved to extended data since this tool was previously published. Also, what do the green triangles (with plus symbols?) signify? And are these two images of the same organism, one showing GFP only and the other showing GFP+dsRed? It is unclear as shown here.

Schematics, maps and nomenclature of promoter regions of Wnt3 (Fig. 3 and c, Ext. Fig 7a) and Sp5 (Figure 3e, Ext. Fig. 7c) need to be more consistent.

Figure 3f, h and 4a: merge the individual plots into single plots; as is, it is difficult to compare each condition.

Schematics in Figures 3a and e: align the pink peaks with the schematic of the gene.

Several graphs lack labeling of the y-axes, e.g. Figure 3a, d, e, 4b, h, I, Extended data Figure 7, 11.

Define labeling and abbreviations, e.g. in Fig. 2g: Hydra1 and Hydra2; Fig. 3g, 4b, c: PP1, PP2, PP3; elsewhere: RLA, Hm-105, HySp5-420, and many more.

For Sp5 ChIP-qPCR: which antibody was used? With what and where is HySp5 tagged?

What do regions labeled PP1, 2 and 3 signify? PCR products for ChIP-qPCR? What is their relationship to pBS?

Line 102: clarify that GSK3 inhibition activates Wnt/b-catenin signaling.

Line 142: does "vertebrate Sp5" refer to ZfSp5? Please clarify.

Figure 4f: what is the meaning of proportionally more binding near the transcriptional start site for HySp5 compared to ZfSp5? Is this a meaningful observation? If only these <1kb binding sites are evaluated, how good is the overlap between HySp5 and ZfSp5? How many genes does this represent?

Figure 4g: what is the relative ranking of these various enriched motifs? Also, indicate p-values for each, as done in other studies.

Line 172-173: the authors suggest that lack of HySp5 binding to the Nanog promoter (Extended data Figure 11c) may be due to the fact that Nanog is absent in the cnidarian genome. However, a more appropriate interpretation is that the Nanog gene is silenced and hence inaccessible in HEK293 cells and active and accessible in mouse ES cells. The authors should clarify this point.

Extended data Figure 9: control injected animals should be shown (i.e. is the amount of phenotype seen for HySp5-337 and HySp5-227 what would be seen in uninjected or control injected embryos?). Also, the apparent synergy between Wnt8 and HySp5 is impressive: the authors should test different concentrations of each mRNA by themselves or in combination.

Reviewer #2:

Remarks to the Author:

In this manuscript, Vogg et al. embark on a series of well thought out and carefully executed experiments in an effort to identify the elusive Head Inhibitor in the freshwater cnidarian Hydra. By taking a comparative approach encompassing 4 species, the authors have identified Sp5 as a novel component of a conserved Wnt signaling feedback loop. The authors demonstrate that Sp5 is a conserved transcriptional repressor of Wnt3 that is positively regulated by Wnt3/ β -catenin in Hydra and mammalian cells. In addition, the authors show that HySp5 and ZfSp5a both bind a similar core consensus motif in HEK293T cells, target a highly overlapping gene set, and regulate Wnt signaling components in particular. Finally, the authors present evidence that Sp5 may be an inhibitor of head organization in Hydra, presenting the most promising candidate to date for the head inhibitor hypothesized by the Geirer-Meinhardt model.

The work presented here is both novel and of broad importance and should be of great interest to the scientific community at large. The wide range of species utilized in the experimental design creates a strong argument for an evolutionarily conserved function for Sp5 and expands upon the previous literature, which has predominantly utilized Hydra. However, the manuscript would benefit from a more careful explanation of the Geirer-Meinhardt model and the motivating context for the work. Additionally, if the author's wish to conclude that the hydra head inhibitor has been identified, this assertion requires more rigorous testing and explanation (see criticisms below).

Major Concerns:

1. Additional description of the Geirer-Meinhardt model should be presented in the introduction to this article. Presently, it is not clear that the model is a primary motivation for the study design and is depicted in Figure 1b without adequate description in the text. Despite the space constraints of the article format, some effort should be made to ensure that a reader outside of the Hydra field can adequately understand the predictions of the model and appreciate the presented findings.

2. Based on the images provided in Figure 1 and extended data figure 4, HySp5 expression does not appear to be expressed in an extended apical to basal gradient at homeostasis, but is rather strongly localized to the head region or the base of the tentacles (Fig 1g). Moreover, expansion of the Sp5 expression zone towards the basal foot appears more robust following tail amputation than following head amputation (Extended Data Fig. 4). This observed expression pattern does not seem to fit with the authors' description of Sp5 expression patterns in Lines 52-59, and the manuscript would benefit from clarification and explanation of how this expression pattern at homeostasis and during regeneration compares with that predicted for a Hydra head inhibitor.

3. The image shown for morphology and Wnt3 expression in head-regenerating animals after mid-gastric bisection following HySp5 siRNA exposure (Figure 2C) appears to show several ectopic heads forming at the apical end of the animal (normal head region) rather than the highly branched morphology that is observed during the emergence of the Sp5 phenotype during

homeostasis. An extended data figure providing additional biological replicates (similar to extended data figure 6) should be provided. Moreover, reasons why the homeostatic phenotype and head regeneration phenotype might differ should be included in the discussion.

4. The authors present compelling evidence that Sp5 is a promising candidate for the hydra head inhibitor. It is both a transcriptional repressor of Wnt3 and positively regulated by Wnt3/ β -catenin in multiple species, and its inhibition results in the formation of new Wnt3-positive bud sites. However, the evidence they present may not be strong enough to support the conclusion that “This study solves a long-standing question, the identification of the Hydra head inhibitor” (Lines 174-175). For instance, Sp5 could alternatively function as an inhibitor of budding or could be co-regulated with additional unknown factors that are also hydra head inhibitors. These conclusions could be rephrased to address this concern. Alternatively, the authors could more rigorously test this assertion. For instance, the authors should better characterize the ‘buds’ formed in HySp5 RNAi animals (integration with gut, ability to eat/digest shrimp, additional tissue markers) to confirm that they do indeed result from axis duplication. One way to further test the assertion would be to perform tissue dissociation/self-organization experiments with HySp5 RNAi or, alternatively, explain why such an experiment was not feasible.

Minor Concerns:

1. Though the authors present strong quantitative evidence the HySp5 regulates Wnt3 activity in HEK293T cells, there is limited testing of this model in Hydra. To address this, the authors might consider using the described transgenic Hydra expressing the HyWnt3:GFP-HyAct-dsRed construct to attempt to quantify the effect of HySp5 RNAi on Wnt3a expression in vivo.

2. Scale bars are needed in Extended Data Figures 4, 6, and 8

Reviewer #3:

Remarks to the Author:

The paper by Vogg et al provides a convincing argument that Sp5 is a repressor of Wnt3 and is activated by Bcatenin during normal homeostasis and regeneration in Hydra. They further make a convincing argument that this process is evolutionary conserved in vertebrates. There are many thorough and well executed experiments. The authors make a very clear case (with some minor comments that I have below) for this regulatory loop, i.e. that the model of Fig 4j is well supported. My main critique is that they have failed to bring this out well as a model to explain how the head organizer region functions and to explain the phenotypes that they observe. Thus if the significance of the work is that they have shown that there is a conserved regulatory loop between bcatenin, Sp5 and Wnt3 in Hydra then I think they have done this well. In my opinion this is not sufficiently significant for publication in this journal. I think the information to provide a model for how this head organizer functions, is contained within their datasets, the authors have just not brought it out thoroughly. In particular this is because they don't adequately show and then explain the spatial distribution of the products. My main comments below are therefore directed at what more is needed to provide a better model. I don't intend that the authors necessarily need to address these comments in detail, but a revision of this manuscript should use their data to better explain how the head is limited to the anterior, why there are many basally located heads in a Sp5 knockdown in contrast to the supernumerary heads in the regenerating polyps.

For instance, how is nuclear bcatenin localized (this may already be published) – throughout the body or only in the head domain. This matters a lot for understanding the context of the rest of the interactions. The ALP treatment seems to increase Sp5 in the basal half of the (e.g. what the authors would term R3, R4 and below), or are levels also increased in the anterior. What does this mean for how Sp5 is normally activated? Its not clear from this experiment what activates Sp5 in

the head region and this is central to understanding the phenotype. Ideally the authors should show a loss of bcatenin function and determine whether Sp5 expression is lost in the head.

In particular in Fig 3d, the authors need to show whether Sp5 is expressed in the very apical – ie can bcatenin drive Sp5 into the apical region even in the presence of wnt3 expression – a higher magnification of the apical domain is needed to assess this important data. Thus the authors show nicely that Sp5 is a target of nbcatenin – but its not at all clear what this means for how the spatial territories are established and hence how the head domain is established.

Does Bcatenin activate wnt3 in just the apical domain or is the model that Wnt3 is activated more broadly and then restricted by Sp5. This is an important component of an understanding of how Sp5 is functioning to limit the head domain. This could be determined by examining wnt3 expression in bcatenin and Sp5 double perturbations.

Also the authors make the case in the intro that the HI should be graded along the apical basal axis, but I'm not sure why this is needed for the model, but this is in part because their model is not clear? But if this is the case the authors need to explain why, and then need to provide better case that this is how Sp5 is expressed. The insitu expression doesn't look at all graded (Fig 1g and 3d – and 3d in particular looks completely uniform apart from the very base) and the quantitative profile should have some statistical analysis to show that expression is indeed graded. Also, what is staining at the very base – is this background?. Finally, the authors should really show that the protein levels are graded to make a case for a graded profile. But in the end its not at all clear anyway why a graded expression is needed to support the HI model.

Given the model that they develop, I don't understand why Wnt3 expression (and the supernumery heads) arise in the basal regions in Sp5 RNAi when Sp5 is not especially expressed in this region (according to their data in Fig 1). Why are there not supernumery heads in the apical domain in non regenerating polyps.. And then why does this differ in the regeneration assay, where heads are now all apical. These supernumery head phenotype are a significant finding of the manuscript but the authors have not explained this phenomena with their model.

In summary, I think that the authors have well justified the regulatory loops shown in figure 4j and that this is conserved. However they have failed to provide of model of what this means for how the head is organized during homeostasis and regeneration. I think the manuscript would be greatly strengthened by providing a model that explains how the head is restricted to the apical pole and how various perturbations, e.g. loss of Sp5, perturbation of bcatenin, regeneration leads to the observed heads phenotypes shown.

The final sentence of the manuscript 181-185 completely misrepresents the significance of the paper – the significance should be the model and how it relates to previous theories that explain the head formation. It has nothing to do with cancer biology – this is a throwaway sentence that weakens the manuscript.

Minor comments

lines 90-91– does not show that Sp5 directly regulates the wnt3 promoter – as loss of DBD from Sp5 could activate or repress an intermediate gene. Experiments of line 94-96 do provide this evidence,.

What is the significance of the expression in gESCs

Reviewer #4:

Remarks to the Author:

The authors provide strong experimental evidence for the evolutionarily conserved nature of a Sp5

and Wnt/ β -catenin interaction across metazoa, factors previously known to interact in vertebrates. They do this by testing Hydra Sp5/Wnt interaction in mammalian cell culture and in the zebrafish embryo. Using these experimental systems they provide strong evidence for autocatalytic activation of HySp5, direct regulation of HySp5 by β -catenin/TCF, direct modulation of wnt through HySp5 and HySp5 interaction with β -catenin and TCF-1. These results are compelling even though these interactions and a graded distribution of the Sp5 gene product are not shown in the context of the animal. Within Hydra they seem to demonstrate Sp5 function by performing RNAi.

This work could be interesting for publication in Nature communications but there are a number of edits to the manuscript and clarifications necessary that I would like to see.

The authors set out to identify the hypothesized diffuse head inhibitor. They did this by screening for genes that are expressed in a graded manner. Even though this approach is valuable it appears that a graded distribution of such an inhibitor may not necessitate graded expression. The identified factor is a transcription factor making it an unlikely candidate for the inhibitor as proposed in the reaction-diffusion system by the Gierer-Meinhardt as the authors acknowledge. Even though a key feature is missing they call it the head inhibitor in the sense of the original model? I do not think this is justified. I would rather suggest that the authors report the findings and discuss them in the light of the existing models for Hydra patterning. They could do so even though they originally set out to identify a diffusible factor. Could a conclusion be that patterning in Hydra does not involve diffusion and one has to rethink the model? I think that the statements made in the introduction and in the concluding paragraph (l. 26, l. 174) are therefore too strong and suggest that they are adjusted accordingly.

Specific comments:

l. 1: Change title to read: evolutionarily conserved

l. 16: Suggested change: Polyps of the cnidarian Hydra maintain..

l. 17: Suggested change: ..., apically the head organizer and basally the foot organizer.

l. 19: This needs rewording: .. through the reactivation underneath the amputation plane of the proper organizer,

l. 21: Here we characterize.. , we identify... (present tense?).

l. 26: Head Inhibitor, change to lower case.

l. 46: The data sets mined (Fig. 1 c(2), c(3), e, f) have been previously discussed by the authors but raw data are not accessible and are not released in the course of this present study. I find this problematic in regards to reproducibility and especially since the foundation of the present study are publicly available data, publicly available data are used to identify Sp5 binding sites in mouse ESCs and also for comparison with genomic occupancies of β -catenin and Sp5 in mammalian cells.

l. 47: add: after 24h of regeneration.

l. 49: this needs re-wording: "unique HI candidate"? It appears Sp5 has some features previously postulated for the head inhibitor.

l. 53: Is there are strong signal in the tentacles as well? The in situ hybridizations suggest so.

l. 55-56: Fig. 1f,g. It appears there is a strong in situ signal in the regenerating foot during the first 24h and that disappears at time point 48h at which a foot has been regenerated? Do these results contradict the gene expression data. Are the authors considering this an unspecific signal?

p. 2. l. 57. "An analysis of cell-type expression by RNA-seq¹⁸ showed that both Sp5 and Wnt3 are predominantly expressed in the gastrodermal epithelial stem cells."

The authors are referring to another dataset that was first presented in 2016. The data have also been used in several studies the raw sequencing data are to date not available. It appears that both the regeneration and tissue specific data sets were created for Hydra vulgaris (strain Jussy)?

This should be stated in the methods section.

I. 63: change: "not from" to "in case of".

I. 66: The terminology should be changed throughout. I do not think "clustered apical cells" is the right way to describe this. Maybe: "Cells of the hypostomal tip", "a distinct population of cells in the apical tip" or "a distinct cluster/group of cells".

I. 66: This phenotype is truly compelling. The authors should provide explicit detail on the electroporation strategy in the methods section or in the supplement including media used, incubation times etc. Did all animals survive this relatively harsh treatment? In previous reports electroporation resulted in a patchy knock down. Is this maybe the reason why new axes form and why they do not find ectopic tentacle formation as in case of Alsterpaulone treatment? I would like to see an in situ hybridization of Sp5 in RNAi treated animals. Are there patches free of Sp5 transcript? Did the authors check via qPCR if Sp5 message was significantly depleted in the RNAi treated animals?

I. 80-82: Do the authors find that axis formation precedes wnt expression? Please clarify. The formation of a new axis in case of budding is indicated by wnt expression even prior to visible evagination (Hobmayer et al. 2000, Fig. 2i). Is this different in Sp5 RNAi animals that form ectopic axes? Maybe the transgenic animals expressing HyWnt3-2149:GFP could be used to combine Sp5 in situ hybridization with GFP immunohistochemistry to further elucidate the interaction.

I. 86: "As the HyWnt3-2149:Luc construct.."? The authors should briefly introduce their approach or rephrase. Maybe add "...using a luciferase reporter assay" at the end of the preceding sentence.

I. 88 – I. 100. These are compelling findings.

I. 111: I suggest to change to "axial patterning".

I. 114: consistent

I. 157: It feels this needs rephrasing. Change to: "These predicted consensus matrixes allowed for the identification of putative Sp5-binding...".

I. 174: The authors claim that they have identified a predicted head inhibitor, a factor for which a diffusive nature has been postulated. I do not think this claim is justified since there is still the possibility of a such a factor with a role in head inhibition. The results are interesting in general and the authors clearly accumulated strong evidence for an evolutionarily conserved interaction that has an important role in Hydra patterning. This alone is worth reporting.

I. 179: I suggest to re-word. It appears that the Sp5 interaction in Hydra development and its evolutionarily conserved nature are previously uncharacterized. The interaction itself appears phylogenetically old and has been characterized in vertebrates.

Figure & Legends

Fig. 4 d): The arrows from gray boxes seem to directly point to FIMO box bypassing green, yellow and purple boxes. I do not think this what the authors intend to show.

Material and Methods

Table 2 (a): spelling: pGEM-T-Easy

I. 283: 3,666ng were used? Please provide a concentration or a volume. What were the plasmids used? What was the concentration of the cells?

I. 309: Please provide cell concentration and volume.

Extended data

Fig. 3: What resources were mined to get Sp/Klf related genes from sponges? Missing genes could be just a function of the reference that was searched? Are there other sponge references that could be consulted? It does not appear to be a thorough phylogenetic analysis of this gene family.

The figure title seems therefore too strong and should be changed. Maybe just provide a statement on similarity and grouping of HySp5 when compared to the available sequences. The last sentences of the figure legend need to be changed accordingly.

- replace "affiliated to" with "of the"
- replace "not affiliated" with "does not group with"?
- in the legend replace "porifers" with "poriferans"
- in the figure legend (Capow, filasterean)

Fig. 7: update to read "(see supplementary material and methods)".

Fig. 8: add ""(see supplementary material and methods)".

Supplementary Information

Hydra genome assembly:

The authors should provide details on sequencing depth, sequencing effort and general stats such as N50, number of scaffolds, contigs. This is valuable information. Did Hydra viridissima animals contain symbionts?

Multiple sequence alignment and phylogenetic analysis:

- Spelling: deuterostomes.
- In the legend of the corresponding figure the authors mention a MAFFT alignment. This contradicts what is stated in the methods.

We thank all reviewers for their constructive comments that helped improve this manuscript significantly. Please find below a point-by-point response to each comment.

Reviewer #1

In this manuscript, the authors provide evidence for Sp5 being a head inhibitor in Hydra. This work confirms prior work in other organisms (zebrafish, mice and humans) that Sp5 is a Wnt target gene that is activated robustly upon Wnt pathway activation. Sp5 in turn acts to repress other critical Wnt target genes, including Wnt3, thereby serving as a negative feedback regulator of Wnt. The authors also provide evidence that Sp5 activates its own expression, a finding consistent with previously published work in mouse ES cells. Overall, this is an interesting study that addresses an important question of how the Hydra head organizer is regulated by interactions between an activator, Wnt3, and an inhibitor, Sp5.

As detailed below, this study has several issues that need to be addressed. More importantly, this study fails to put to rest a discrepancy in the analysis of Sp5 acting downstream of Wnt/b-catenin signaling: is Sp5 primarily a transcriptional repressor, a transcriptional activator, or both? The authors provide evidence that hySp5 can act as both, a repressor of such genes as Wnt3 and an activator of itself. How Sp5 achieves this dual function is unclear, and the literature provides conflicting results: Fujimora et al. (2007, PMID: 17090534) and Huggins et al. (2017, PMID: 29044119) provide evidence that Sp5 acts primarily as a repressor. In contrast, in mice, Kennedy et al. (2016, PMID: 26969725) argued that Sp5 acts to activate Wnt/b-catenin target gene expression. This discrepancy can certainly be rationalized many different ways. However, it would be preferred if the authors of this paper could address this question head on and possibly resolve at least Sp5's ancestral/original role as a repressor or activator or both. If it is in fact the case that Sp5 acts as both repressor and activator, it would be important to address how this happens. Interaction between Sp5 and b-catenin/Tcf has been shown by others and this could lead to either inhibition or activation. However, this fails to explain how Sp5 could achieve both functions as a repressor and activator. Something is missing and it would be good if the authors could shed some light on this.

M.V. et al: To answer this important question we have produced new data in *Hydra* (**Figure 3**) as well as in HEK293T cells (**Figure 6**). In *Hydra* we have added numerous Sp5 loss-of-function assays when the activity of the Wnt/ β -catenin signaling is either physiological, or inhibited by β -catenin(RNAi) (NEW) or enhanced upon Alsterpaullone treatment (NEW) in three main contexts: homeostatic, regenerative or reaggregation (NEW, **Figure 3d, 3e**, see also summary Table-1 below). All these results converge to demonstrate the repressor activity of Sp5 on *Wnt3* expression, and subsequently on Wnt/ β -catenin signaling *in vivo*.

	Sp5 (RNAi) + Wnt/ β -catenin signaling activity			
In Hydra	physiological	\downarrow β -cat (RNAi)	\uparrow ALP treatment	\downarrow β -cat (RNAi) + \uparrow ALP
homeostatic	Fig 2a ¹ , b ² , d ¹ , e ³ Suppl Fig. 6 ¹ Suppl Fig 12 ^{1,2,4}	Figure 3a ¹ Suppl Fig. 9 ¹	Figure 3b ¹ , c ² Fig. 5a ² Suppl Fig. 14 ^{1,2}	Figure 3c ² Suppl Fig. 10 ^{1,2}
regeneration	Figure 2c ^{1,2} Suppl Fig. 4 ² Suppl Fig. 7 ^{1,2}	Suppl Fig. 8 ¹		
reaggregation			Suppl Fig. 11 ^{1,2}	Figure 3d ^{1,2} , 3e ² Suppl Fig. 11 ^{1,2}

Summary Table-1 summing up the different contexts where the function of HySp5 was tested in *Hydra* through RNAi loss of function experiments. Output of these experiments are morphological changes¹, gene expression levels² (*Wnt3*, *Sp5*, *HyBra1*, *Tsp1*, *Kazal1*, ...), RFamide patterns³, HyWnt3:GFP reporter construct activity⁴. When written black, the Figure contains useful control experiments but no *Sp5*(RNAi) experiments.

In the human HEK293T cells, we have further tested and compared how HySp5 and ZfSp5a act as activators and/or repressors of transcription. In cells overexpressing these factors, we had previously shown that both HySp5 and ZfSp5a repress transcription driven by the *Hydra* and zebrafish *Wnt3* promoter, respectively (**Figure 4**). We have now generated massive RNA-seq data from such cells. We only considered the gene modulations that are induced through DNA-binding, i.e. induced by the full-length HySp5 and ZfSp5a proteins but not by their truncated versions lacking the DNA-binding domain. We found that 99.4% of the genes regulated by HySp5 are also regulated by ZfSp5a (13'158/13'251) while an additional series of 5'441 genes is regulated exclusively by ZfSp5a (**NEW data shown in Figure 6 panels a, b, c, d, e**). When we crossed the RNA-seq data with the ChIP-seq data sets we previously produced, we identified 349 genes that appear directly repressed by one or the other factor (236 by HySp5, 196 by ZfSp5a). Among these 83 (24%) are repressed by the two Sp5 orthologs. In contrast, only 165 genes were found directly up-regulated, mostly by ZfSp5a (142 by ZfSp5a, 28 by HySp5) and only 5 (3%) in common between both Sp5 orthologs (**Figure 6, panel i**). Additionally, we demonstrate that HySp5 interacts with β -catenin and TCF (**Figure 5, panel f**). By using ChIP-seq analysis we also found that both orthologs bind largely overlapping sets of proximal regulatory regions (**Figure 6, panels a-e**) and recognize the same consensus binding side (**Figure 6, panel f**). However, ZfSp5a readily binds to more upstream regulatory regions (> 10 kb from TSS) and TBX/SOX consensus sites were also found enriched in enhancers bound by ZfSp5a, but not in those bound by HySp5. Tbx and Sox transcription factors positively regulate gene expression by interacting with β -catenin and Sp1 (*Zorn et al., Mol. Cell 1999; Chen et al., PNAS 2006*). Thus, even though *Hydra* and vertebrate Sp5 proteins can both activate gene expression through interaction with β -catenin, Sp5 may have evolved after Cnidaria divergence the capacity to interact with a wider set of transcription factors, such as TBX/SOX

transcription factors and/or may have acquired the capacity to recognize a consensus motif enriched in long-range enhancers to regulate gene expression.

In HEK293T cells	HySp5	ZfSp5a	Human β -cat
Wnt3-reporter constructs Wnt3 promoter Sp5-BS	Fig. 4a, b, c Suppl Fig 13a	Fig. 4d, e Suppl Fig 13b	Fig. 4b, c, e
ChIP-seq data: Wnt3 prom		Fig. 4d, f	
Sp5-reporter constructs Sp5 promoter Sp5-BS	Fig. 5e Suppl Fig 13c		Fig. 5c, e
ChIP-seq data: Sp5 prom	Fig. 5b, d		
ChIP-seq data: genome wide DNA-bound sites	Fig. 6a, b, c, d, e, f Suppl Fig 16a, b, c	Fig. 6a, b, c, d, e, f Suppl Fig 16b, c	
RNA-seq data: genes up- and down- regulated	Fig. 6a, g, h	Fig. 6a, g, h	
RNA-seq + ChIP-seq data: direct Sp5 regulation	Fig. 6i	Fig. 6i	
Co-IP data: Sp5 interacts with β-catenin and TCF	Fig. 5f Suppl Fig 15b, c		

Summary Table-2 summing up the different contexts where the functions of Sp5 from *Hydra* and zebrafish were tested in HEK293T cells

We also clarify in this revised version the critical link between Sp5 repressor function and Wnt/ β -catenin signaling. Although β -catenin efficiently translocates to the nucleus upon Wnt signaling stimulation in HEK293T cells (*Tan et al., Plos One 2012; Tan et al., BMC Syst Biol 2014*), we observed only marginal transcriptional changes upon expression of a constitutive active form of β -catenin (in agreement with the studies by *Li et al., PNAS 2011; Huggins et al., Nature Commun 2017*). This complicates the identification of genes that would be activated by Sp5/ β -catenin transcriptional complexes, and thus the comparison of the activity of HySp5 and ZfSp5a in this model. However, our results suggest that the cnidarian and vertebrate Sp5 proteins, which both bind to proximal regulatory elements located in the vicinity of the transcriptional start sites, have similar repressive activities. By contrast, if we take the number of shared up-regulated genes into account, the activator function appears much less conserved across evolution. Distal regulatory elements bound by ZfSp5a are also enriched in consensus motifs for transcription factor binding sites not found in the HySp5 bound elements, such as TBX/Sox. It is possible that part of the functions of vertebrate Sp5 have evolved through the acquisition of new transcriptional partners.

All together these results that combine the measurements of reporter constructs activities, DNA-binding activity and transcriptional regulations when HySp5 and ZfSp5 are expressed in HEK293T cells (See Summary Table-2 just above), shed light onto the

ancestral function of HySp5 and allowed us to propose a series of definitive conclusions on the repressor and activator roles of Sp5 across evolution:

- HySp5 and ZfSp5 similarly repress the activity of reporter constructs driven by the *Hydra* and zebrafish *Wnt3* promoter respectively (**Figure 4**)
- HySp5 and ZfSp5 regulate the expression of a largely overlapping set of target genes
- HySp5 and ZfSp5 down-regulate the expression of an overlapping (24%) series of genes in HEK293T cells (**Figure 6**)
- HySp5 and ZfSp5 both bind to the regulatory regions of genes that are involved in cell surface receptor signaling, Wnt signaling pathway, ... and negative regulation of transcription via RNA polymerase II (**Supplementary Figure 16**)
- **The repressor function of Sp5 on Wnt/ β -catenin signaling appears to be widely conserved across eumetazoans, possibly representing the ancestral function of Sp5**
- ZfSp5a binds to a larger number of target genes
- ZfSp5a binds to sequences that are not recognized by HySp5, namely TBX/SOX consensus sites
- The activator function of HySp5 appears much more limited than that of ZfSp5a
- **The activator function of Sp5 largely evolved after Cnidaria divergence, possibly through the recruitment of novel co-transcriptional partners and/or novel types of regulatory sequences.**

Consistent with Sp5 being a repressor, the authors show that Sp5 knockdown (KD) by siRNA leads to ectopic Wnt3 expression. Here, the authors should test whether Sp5 expression is elevated or diminished upon Sp5 KD. Knockout of SP5 in human ES cells led to elevated expression of SP5, indicating that SP5 served to repress its own expression. A similar experiment in Hydra with siRNA KD may be difficult to examine by in situ hybridization for Sp5. Instead, the authors could test this with a transgenic Hydra carrying a Sp5 reporter, similar to the experiments shown in Fig. 2f and g with the HyWnt3:GFP animals. This experiment would require using the HySp5-2992 reporter (e.g. GFP) in Hydra and showing that this reporter is either activated (indicating that Sp5 is a repressor of itself) or inhibited (indicating that Sp5 is an activator of itself). Currently, the authors only provide evidence for Sp5 being an activator in a HEK293 reporter assay with overexpression of tagged HySp5.

M.V. et al: We thank for suggesting this interesting experiment, however technical limitations currently prevent us from answering this question. We would like to stress that generating a transgenic line in *Hydra* can take several weeks to more than a year. In the present case, we have prepared this *HySp5* reporter construct and despite sustained efforts (927 injected eggs, 23 hatched *Hydra*, 4 GFP positive babies), we have not being able to obtain a stable line. Concerning the Sp5 autoregulation, we have tested by ISH and qPCR the expression of *HySp5* after its knockdown and we observed either a similar

level or an up-regulation of Sp5 as shown for human SP5 in ES cells (*Huggins et al., Nature Commun. 2017*). However, we believe that *HySp5* regulation in response to its knockdown is complex as highly regulated *in time and space*. *HySp5* knockdown likely leads to an immediate de-repression of *Wnt3* and an activation of *Wnt3/β-catenin* signaling, which in turn induces the expression of *HySp5*. Alternatively, *HySp5* might have a repressing effect on its own promoter as observed in human ES cells. However, without having the Sp5 reporter line in hands to precisely monitor the expression of *Sp5* in live animals, we can currently not convincingly answer this question. Nevertheless, we revised our model, which now shows that *HySp5* might also have a repressing effect on its own promoter.

Figure 3a-c: The ~300 fold increase in basal expression levels of the HyWnt3-1763 reporter compared to the HyWn3-2149 reporter is surprising and unexplained. Do the authors think that this increase in expression is entirely due to loss of Sp5 binding sites, resulting in de-repression? This possibility should be explored further. The more refined deletions of this region indicate that the repressive elements reside in two ~100bp regions, one at -386/-286 and one at -95/-1. Extended Figure 7a indicates two predicted Sp5 binding sites reside in these regions (unfortunately, as illustrated, it is not possible to correlate the map shown in Ext.Fig.7a with the promoter elements shown in Figures 3a and c; this should be fixed); does mutation of these pBSs lead to loss of repression of ΔRep-D3?

M.V. et al: The increased basal activity is likely due to the loss of the 386 bp *HyWnt3* repressor, which has been described to repress *Wnt3* transcription (*Nakamura et al., PNAS 2011*). Within this repressor, we have identified three candidate Sp5 binding sites, two of them falling in the two regions we have identified as necessary for repression (Figure 4a, 4c). As the referee correctly states, testing whether these sites are entirely responsible for *Wnt3* repression by Sp5 requires mutagenizing these sequences *in vivo*. Ideally, CRISPR mediated mutagenesis of the predicted Sp5 binding sites *in vivo* would represent the best experimental approach to address this question but these experiments would require the generation of mutant animals, something not doable within the revision time period of this study.

On line 117, the authors state “To test whether HySp5 autoactivates its own promoter, we performed ChIP-qPCR experiments...” Such an experiment does not test whether Sp5 autoactivates; it shows that Sp5 can bind in the promoter of Sp5; whether Sp5 activates or inhibits is a separate question that is addressed in Fig. 3h.

M.V. et al: We agree and we revised the text accordingly.

The interpretation of Fig. 3h is a bit flawed: the authors claim that (line 122) “HySp5....increases activity of the HySp5 promoter, an effect enhanced upon b-catenin

overexpression.” This claim is not supported by the data: the level of trans-activation between HySp5 alone and HySp5+b-catenin are essentially the same, i.e. there is no enhancement of activity in the presence of b-catenin. It is also a little misleading and confusing that the authors split the data into 2 graphs: these should be combined into one single graph.

M.V. et al: We followed this advice and pooled the data. The new graph is now shown in Figure 5e. By calculating the fold change of luciferase activity, we show that β -catenin mildly but significantly enhances the activating effect of HySp5 on the *HySp5* promoter.

Figure 3i is unclear: what does inclusion of HySp5-2992:Luc (indicated by “+”) mean? Does this co-IP require the presence of this reporter element? Will the interactions between Sp5 and b-catenin/Tcf1 be observed if this reporter element were not present? I assume so, but it is unclear why the figure is labeled as is. Also in this figure, it is unclear whether Sp5 co-immunoprecipitates with endogenous b-catenin or with the slightly truncated transfected version of b-catenin

M.V. et al: We understand the concern raised by Reviewer 1 and we have performed additional Co-IP experiments without transfecting the *HySp5-2992:Luc* construct. We now show in **Figure 5f** that that the observed interactions between HySp5 and β -catenin/TCF is, as expected, observed in the absence of the reporter construct. It is very difficult to separate the endogenous from the truncated β -catenin as the bands run very close together (95 vs 92kD). Even though we ran the Co-IP on a low-percentage SDS gel, we could not separate the two bands more than shown in **Fig. 5f** and **Supplementary Fig. 15b**. In **Figure 5f** we show that Sp5 co-immunoprecipitates with truncated β -catenin, however on membranes exposed for longer time (as shown in **Supplementary Fig. 15b**), we see that Sp5 interacts with endogenous β -catenin. Truncated β -catenin is labeled with a black arrow and the endogenous protein with a white arrow.

Figure 1e: what do the numbers in the third column (under “Apical, R1, R2, R3, R4, Basal”) signify? Are these RPKMs?

M.V. et al: These numbers indicate the mean value of the number of reads measured in three biological replicates. This information is now added in the legend of **Figure 1e**.

Figure 2f could be moved to extended data since this tool was previously published. Also, what do the green triangles (with plus symbols?) signify? And are these two images of the same organism, one showing GFP only and the other showing GFP+dsRed? It is unclear as shown here.

M.V. et al: We agree with the suggestion of reviewer 1 and we moved these images to **Supplementary Fig. 12a-b**. In **Supplementary Fig. 12a** we show images of the same animal. The ++ and + symbols indicate maximal and intermediate levels of GFP, respectively. We revised the figure legends accordingly.

Schematics, maps and nomenclature of promoter regions of Wnt3 (Fig. 3 and c, Ext. Fig 7a) and Sp5 (Figure 3e, Ext. Fig. 7c) need to be more consistent.

M.V. et al: We revised our maps and added e.g. the TCF binding sites shown for the *Wnt3* and *Sp5* promoters in **Figure 4a** and **Figure 5b** to the **Supplementary Fig. 13**.

Figure 3f, h and 4a: merge the individual plots into single plots; as is, it is difficult to compare each condition.

M.V. et al: We merged the data of the previous **Figure 3f-h** and we now present the data in **Figure 5c**. We prefer to present the zebrafish *Sp5* data in two individual plots as we did not test a construct for *ZfSp5I1* that lacks the DNA binding domain.

Schematics in Figures 3a and e: align the pink peaks with the schematic of the gene.

M.V. et al: We revised the figures accordingly.

Several graphs lack labeling of the y-axes, e.g. Figure 3a, d, e, 4b, h, I, Extended data Figure 7, 11.

M.V. et al: Previous **Figure 3d**, now shown in **Figure 5a** shows the relative *HySp5* expression level as written above the graph. We added the labeling to Figure 4a, 5b, and Supplementary Figures 13, 15a, 16a-b.

Define labeling and abbreviations, e.g. in Fig. 2g: Hydra1 and Hydra2; Fig. 3g, 4b, c: PP1, PP2, PP3; elsewhere: RLA, Hm-105, HySp5-420, and many more.

M.V. et al: Thanks for pointing this out. We revised the labeling and defined the abbreviations used in the Figures.

For Sp5 ChIP-qPCR: which antibody was used? With what and where is HySp5 tagged?

M.V. et al: We added this information to the materials and methods section.

What do regions labeled PP1, 2 and 3 signify? PCR products for ChIP-qPCR? What is their relationship to pBS?

M.V. et al: PP means Primer Pair. We added this information to the corresponding figure legends.

Line 102: clarify that GSK3 inhibition activates Wnt/ β -catenin signaling.

M.V. et al: We revised the text accordingly.

Line 142: does “vertebrate Sp5” refer to ZfSp5? Please clarify.

M.V. et al: Yes, indeed we infer that the data obtained with ZfSp5a provides a good representation of a generic “vertebrate Sp5”. We have revised the text accordingly.

Figure 4f: what is the meaning of proportionally more binding near the transcriptional start site for HySp5 compared to ZfSp5? Is this a meaningful observation? If only these <1kb binding sites are evaluated, how good is the overlap between HySp5 and ZfSp5? How many genes does this represent?

M.V. et al: We have now confirmed these results by analyzing independent biological replicates of ChIP-seq. This analysis confirms the differential distribution of HySp5 and ZfSp5a bound elements. The same differential distribution is observed when sites located at < than 1kb from the gene TSS are observed (see figure below, panel a). Considering only HySp5/ZfSp5a bound elements within ≤ 1 kb from the gene TSS, the number of genes assigned to these elements is 9248 and 12033, respectively. We have now provided a more detailed description of the distribution of HySp5/ZfSp5a bound elements and their density in different portions of the gene regulatory domains in **Figure 6d-e**. In this analysis we have considered the 5kb upstream of the gene TSS as the proximal gene regulatory domain. Whether the differential distribution of HySp5/ZfSp5a bound elements constitutes a meaningful biological observation is an interesting point. To address this question, we performed RNA-seq experiments in HEK293T cells transfected either with HySp5 or ZfSp5a and crossed this information with our ChIP-seq data to identify genes directly repressed or activated by these proteins. This is now described in **Figure 6g-i**. We have also compared the consensus motifs enriched in HySp5 and ZfSp5a bound elements. These data are presented in **Figure 6f**. We provide below a more detailed analysis of the TF binding sites enriched in HySp5 and ZfSp5a bound elements located in the proximal regulatory region of the genes (<5kb from the TSS) or at more distal locations. The conclusions of these analyses are that although both HySp5 and ZfSp5a have similar repressive capacities, they differ in their ability to activate endogenous HEK293T target genes. We also observed that HySp5 and ZfSp5a bound elements are differentially enriched in sets of consensus motifs for different TFs. Particularly, elements specifically bound by ZfSp5a, and located in proximal and distal regulatory regions, display differential enrichment of some TF consensus sequences (see panels b and c of the figure below). However, we could not find a clear correlation between these enrichments and positive or negative transcriptional responses of the

assigned transcriptional target genes. Although these results suggest that HySp5 and ZfSp5a may differentially interact with a number of other transcriptional co-regulators, their experimental validation would require a large and complex set of experiments that are beyond the scope of this study. Because of this, we chose to present a simplified version of these results, limited to what is shown in **Figure 6**.

Figure-1. (a) Bar plot representing the percentage of Sp5 bound elements at different distances from the assigned gene TSS for HySp5 (black) or ZfSp5a (grey). **(b)** Bar graph representing the percentage of elements specifically bound by HySp5 or ZfSp5a that are located either in the 5kb immediately upstream of the gene TSS or at longer genomic distances. Note that while HySp5 specific elements are equally distributed among these locations, the majority of ZfSp5 specific elements are found at mid-long distances from the gene body. **(c)** Heat map plot representing the distribution and the $-\log_{10}(E)$ value of enriched motifs associated with known TF consensus sequences identified in HySp5 and/or ZfSp5a bound elements. Regions bound specifically by HySp5, ZfSp5a or by both proteins were identified using MACS2. The sites were classified based on their location either in the 5kb immediately upstream of the gene TSS or at larger genomic intervals. Elements with the lowest MACS2 score (Q25 quartile of each dataset) were discarded from the analysis. For each set of sequences, enriched motifs were identified using the MEME ChIP suite. The enriched consensus sequences were matched with matrixes of known TFs using the TOMTOM tool of the same suite. For each genomic location, the matched TF consensus motifs were plotted according to the E value calculated by the DREME or MEME tools of the MEME ChIP suite. Pale pink colors represent high E values, indicating lowly enriched motifs. Dark green color represent highly relevant consensus binding sites (low E values). Note that for both proximal and distal HySp5/ZfSp5a bound elements the more significant consensus motifs belong to the Zinc finger transcription factor ZNF263 and to the different Sp/Klf family members. Apart from these

consensus sequences, proximal and distal regulatory elements are differentially enriched for a number of consensus TF consensus motifs (highlighted by magenta boxes). Furthermore ZfSp5a bound elements located at >5Kb from the gene TSS are enriched in Sox13, EWSR-FLI1 and Klf4 consensus motifs, not or lowly enriched in any other element. Instead, proximal ZfSp5 bound elements are enriched in a larger variety of TF consensus sequences including those of Tbx, Pou and different Klf family members.

Figure 4g: what is the relative ranking of these various enriched motifs? Also, indicate p-values for each, as done in other studies.

M.V. et al: We now include in **Figure 6f** the top 5 enriched motifs identified in HySp5 and ZfSp5 bound elements, ordered according to their ranking and showing the E value calculated by the MEME/DREME (as indicated).

Line 172-173: the authors suggest that lack of HySp5 binding to the Nanog promoter (Extended data Figure 11c) may be due to the fact that Nanog is absent in the cnidarian genome. However, a more appropriate interpretation is that the Nanog gene is silenced and hence inaccessible in HEK293 cells and active and accessible in mouse ES cells. The authors should clarify this point.

M.V. et al: We clarified this point in the manuscript.

Extended data Figure 9: control injected animals should be shown (i.e. is the amount of phenotype seen for HySp5-337 and HySp5-227 what would be seen in uninjected or control injected embryos?). Also, the apparent synergy between Wnt8 and HySp5 is impressive: the authors should test different concentrations of each mRNA by themselves or in combination.

M.V. et al: We tested different concentrations of each mRNA or in combination and we now provide this information in Extended Data 3 together with the information of control-injected embryos.

Reviewer #2 (Remarks to the Author):

In this manuscript, Vogg et al. embark on a series of well thought out and carefully executed experiments in an effort to identify the elusive Head Inhibitor in the freshwater cnidarian Hydra. By taking a comparative approach encompassing 4 species, the authors have identified Sp5 as a novel component of a conserved Wnt signaling feedback loop. The authors demonstrate that Sp5 is a conserved transcriptional repressor of Wnt3 that is positively regulated by Wnt3/ β -catenin in Hydra and mammalian cells. In addition, the authors show that HySp5 and ZfSp5a both bind a similar core consensus motif in HEK293T cells, target a highly overlapping gene set, and regulate Wnt signaling components in particular. Finally, the authors present evidence that Sp5 may be an inhibitor of head organization in Hydra, presenting the most promising candidate to date for the head inhibitor hypothesized by the Gierer-Meinhardt model.

The work presented here is both novel and of broad importance and should be of great interest to the scientific community at large. The wide range of species utilized in the experimental design creates a strong argument for an evolutionarily conserved function for Sp5 and expands upon the previous literature, which has predominantly utilized Hydra. However, the manuscript would benefit from a more careful explanation of the Gierer-Meinhardt model and the motivating context for the work. Additionally, if the author's wish to conclude that the hydra head inhibitor has been identified, this assertion requires more rigorous testing and explanation (see criticisms below).

Major Concerns:

1. Additional description of the Geierer-Meinhardt model should be presented in the introduction to this article. Presently, it is not clear that the model is a primary motivation for the study design and is depicted in Figure 1b without adequate description in the text. Despite the space constraints of the article format, some effort should be made to ensure that a reader outside of the Hydra field can adequately understand the predictions of the model and appreciate the presented findings.

M.V. et al: We agree with this comment and we added a paragraph about the key points of the Meinhardt-Gierer model in the introduction as follows:

"Gierer and Meinhardt used the results obtained from a series of transplantation experiments to propose a general mathematical model of morphogenesis¹⁶. Their model revisits the Turing model, which is based on the reaction-diffusion model where two substances that exhibit distinct diffusion properties and interact with each other, form a minimal regulatory loop that suffices for de novo pattern formation¹⁷. Gierer and Meinhardt refined this model by posing that the activation component acts over short-range distance, while the inhibition one acts over long-range distance, and by distinguishing between "effective concentrations of activator and inhibitor, on one hand, and the density of their sources on the other"¹⁶. These models proved to efficiently simulate basic

properties of pattern formation and were validated by molecular data in a variety of developmental contexts¹⁸”.

2. Based on the images provided in Figure 1 and extended data figure 4, HySp5 expression does not appear to be expressed in an extended apical to basal gradient at homeostasis, but is rather strongly localized to the head region or the base of the tentacles (Fig 1g). Moreover, expansion of the Sp5 expression zone towards the basal foot appears more robust following tail amputation than following head amputation (Extended Data Fig. 4). This observed expression pattern does not seem to fit with the authors' description of Sp5 expression patterns in Lines 52-59, and the manuscript would benefit from clarification and explanation of how this expression pattern at homeostasis and during regeneration compares with that predicted for a Hydra head inhibitor.

M.V. et al: Following amputation, Sp5 is expressed in head- and foot-regenerating tips, however the expression is exclusively sustained in head regenerating tips. An injury can trigger the up-regulation of genes at both wound sites but many of them are sustained at only one side (Technau et al. Development 1999; Wenger et al., Seminars in Immunology 2014), as it is the case for Sp5. To rule out a possible role for Sp5 in foot regeneration we knocked-down Sp5 in foot-regenerating animals and we can now demonstrate that all Sp5(RNAi) animals regenerate their foot normally (NEW DATA, **Supplementary Fig. 7d**). Thus, the transient up-regulation of Sp5 in foot regenerating polyps is most likely linked to injury signals without any role in foot regeneration.

We agree that it is difficult to appreciate an apical-to-basal graded expression of Sp5 in intact animals with the whole-mount ISH (**Fig. 1g and Supplementary Fig. 4**). Throughout the manuscript, we now say that Sp5 is predominantly expressed in the head. Nevertheless, we also noted that the Sp5 expression pattern varies slightly from animal to animal (see in **Supplementary Fig. 4c** foot-regenerating animals at 24 hpa or in **Supplementary Fig. 4b** head-regenerating animals at 12 hpa). Thus, we suspect that the expression of Sp5 is actually oscillating in homeostatic conditions, a point that is more explicitly discussed in the discussion section. (see the section: **The Sp5 RNAi phenotype suggests an oscillatory regulation for Sp5**). In addition, we do believe that one can appreciate more easily a graded expression in budding (not shown) or in regenerating animals (see for example the head-regenerating animals in **Supplementary Fig. 4b**), suggesting that the apical-to-basal distribution of Sp5 transcripts is more stable in developing animals.

3. The image shown for morphology and Wnt3 expression in head-regenerating animals after mid-gastric bisection following HySp5 siRNA exposure (Figure 2C) appears to show several ectopic heads forming at the apical end of the animal (normal head region) rather than the highly branched morphology that is observed during the emergence of the Sp5 phenotype during homeostasis. An extended data figure providing additional biological replicates (similar to extended data figure 6) should be provided. Moreover,

reasons why the homeostatic phenotype and head regeneration phenotype might differ should be included in the discussion.

M.V. et al: We agree with Reviewer-2 that additional biological replicates need to be presented, something we have now done in the NEW **Supplementary Figure 7**. In addition, we now discuss more precisely in the revised version of the manuscript the differences between the *Sp5* phenotype in intact versus regenerating conditions. See the NEW section of the discussion entitled “**The physiological and developmental variations of the Sp5(RNAi) phenotype reflect the relative spatial distribution of Sp5 and Wnt3/ β -catenin signaling activity**”.

4. The authors present compelling evidence that Sp5 is a promising candidate for the hydra head inhibitor. It is both a transcriptional repressor of Wnt3 and positively regulated by Wnt3/ β -catenin in multiple species, and its inhibition results in the formation of new Wnt3-positive bud sites. However, the evidence they present may not be strong enough to support the conclusion that “This study solves a long-standing question, the identification of the Hydra head inhibitor” (Lines 174-175). For instance, Sp5 could alternatively function as an inhibitor of budding or could be co-regulated with additional unknown factors that are also hydra head inhibitors. These conclusions could be rephrased to address this concern. Alternatively, the authors could more rigorously test this assertion. For instance, the authors should better characterize the ‘buds’ formed in HySp5 RNAi animals (integration with gut, ability to eat/digest shrimp, additional tissue markers) to confirm that they do indeed result from axis duplication. One way to further test the assertion would be to perform tissue dissociation/self-organization experiments with HySp5 RNAi or, alternatively, explain why such an experiment was not feasible.

M.V. et al: We fully understand the concern of Reviewer-2 and we removed the sentence “*This study solves a long-standing question, the identification of the Hydra head inhibitor*” from the revised version of our manuscript. In parallel, we have also strengthened the scientific arguments that support *Sp5* as a key head inhibitor in *Hydra*.

- The *Sp5* phenotype occurs first in the budding region, a developmentally competent region and a zone of low *Sp5* expression (**Figure 2a**)
- we further characterized the ectopic heads of *Sp5*(RNAi) animals and we show that they all express three head markers: *Wnt3*, *HyBra1* and *Tsp1* (**Figure 2b**)
- we also show with the *Kazal1* marker (gland cells of the gastric cavity) that the endodermal tissue of the ectopic axis is connected to the parental axis (**Fig. 2b**)
- we show in **Fig. 2e**, **Supplementary Fig. 6d** and **Supplementary Movies 1-4** that these ectopic heads catch preys and are connected to gastric cavities that eat *Artemia*. In other words, they form the complete *Hydra* anatomy
- The *Sp5* phenotype is NOT restricted to the budding zone as when silencing is reinforced with an increasing numbers of siRNA electroporations, ectopic axes appear in the apical half of the animals (**Supplementary Fig. 6**)

- The *Sp5* phenotype is NOT restricted to the budding zone as knocking down *Sp5* in head regenerating conditions triggers the formation of multiple heads from head- but not foot-regenerating tips (**Supplementary Fig. 7**)
- Finally, to demonstrate that *Sp5* acts as a general strong head inhibitor, we performed for the first-time dissociation-reaggregation experiments on RNAi animals. In such reaggregates, the knockdown of *Sp5* triggers multiple head formation, which is characterized by an increased number of *Wnt3*-expressing clusters. To support this data, we also performed reaggregation experiments with *β-catenin*(RNAi) animals, which showed smaller *Wnt3*-expressing clusters and the development of only a few tentacles (**Figure 3d-e and Supplementary Fig. 11**). We are grateful to Reviewer-2 who suggested this excellent experiment

In summary, we believe that we now have strong arguments to exclude that the *Sp5* phenotype is a budding-restricted phenotype.

Minor Concerns:

1. *Though the authors present strong quantitative evidence that HySp5 regulates Wnt3 activity in HEK293T cells, there is limited testing of this model in Hydra. To address this, the authors might consider using the described transgenic Hydra expressing the HyWnt3:GFP-HyAct-dsRed construct to attempt to quantify the effect of HySp5 RNAi on Wnt3a expression in vivo.*

M.V. et al: We agree that in the previous version of the manuscript, there was a limited testing of our model in *Hydra*. We now present in Figure 3 new data, which demonstrates that *Sp5* represses *Wnt3* expression in *Hydra*. In particular, we performed double knockdown experiments to demonstrate that the *Sp5* phenotype requires an active Wnt/ β -catenin signaling (**Fig. 3a and Supplementary Fig. 9**). In addition, we performed RNAi experiments in combination with drug treatments, which resulted in an increase in *Wnt3* expression along the body axis after *Sp5* knockdown (**Fig. 3b-c and Supplementary Fig. 10**). Finally, and as described above we performed reaggregation experiments and similarly as intact and head-regenerating *Sp5*(RNAi) animals, aggregates knocked-down for *Sp5* were multi-headed (**Fig. 3d-e and Supplementary Fig. 11**).

2. *Scale bars are needed in Extended Data Figures 4, 6, and 8*

M.V. et al: We revised the figures accordingly and we have added the missing scale bars.

Reviewer #3

The paper by Vogg et provides a convincing argument that Sp5 is a repressor of Wnt3 and is activated by β -catenin during normal homeostasis and regeneration in Hydra. They further make a convincing argument that this process is evolutionary conserved in vertebrates. There are many thorough and well executed experiments. The authors make a very clear case (with some minor comments that I have below) for this regulatory loop, i.e. that the model of Fig 4j is well supported. My main critique is that they have failed to bring this out well as a model to explain how the head organizer region functions and to explain the phenotypes that they observe. Thus if the significance of the work is that they have shown that there is a conserved regulatory loop between β -catenin, Sp5 and Wnt3 in Hydra then I think they have done this well. In my opinion this is not sufficiently significant for publication in this journal. I think the information to provide a model for how this head organizer functions, is contained within their datasets, the authors have just not brought it out thoroughly. In particular this is because they don't adequately show and then explain the spatial distribution of the products. My main comments below are therefore directed at what more is needed to provide a better model. I don't intend that the authors necessarily need to address these comments in detail, but a revision of this manuscript should use their data to better explain how the head is limited to the anterior, why there are many basally located heads in a Sp5 knockdown in contrast to the supernumerary heads in the regenerating polyps.

For instance, how is nuclear β -catenin localized (this may already be published) – throughout the body or only in the head domain. This matters a lot for understanding the context of the rest of the interactions. The ALP treatment seems to increase Sp5 in the basal half of the Hydra (e.g. what the authors would term R3, R4 and below), or are levels also increased in the anterior. What does this mean for how Sp5 is normally activated? Its not clear from this experiment what activates Sp5 in the head region and this is central to understanding the phenotype. Ideally the authors should show a loss of bcatenin function and determine whether Sp5 expression is lost in the head.

M.V. et al: We understand the concerns of Reviewer-3 and we have performed loss of function of β -catenin to bring new evidences about our head organizer model. Indeed previous studies have analyzed the spatial and cellular distribution of β -catenin: *Hydra β -catenin* is uniformly expressed throughout intact polyps (Gee et al., *Developmental Biology* 2010; Iachetta et al., *Int J Dev Biol* 2018). However, Broun et al (*Development* 2005) who used an anti- β -catenin antibody (unfortunately no longer available) found β -catenin predominantly nuclear in cells of the hypostome compared to cells of the body column where β -catenin is localized in the cytoplasm and membranes. They treated *Hydra* with the GSK3 β inhibitor Alsterpaullone (ALP) and could increase the level of nuclear β -catenin along the body column, preceding the appearance of ectopic tentacles, interpreted as ectopic head organizers. Their conclusion is that clusters of cells where β -catenin is nuclear are necessary for head organizer formation.

To follow this line, we knocked-down β -catenin and quantified the expression of β -catenin in head and body column tissue. While we could significantly down-regulate the expression in body column tissue (**Supplementary Fig. 8a**), we did not succeed in silencing β -catenin in head tissue. To circumvent this problem, we quantified the expression of *Sp5* after ALP treatment in head and body column tissue at two different time points (**Supplementary Fig. 14b**). Consistently with our WISH data (**Fig. 5a and Supplementary Fig. 14a**), ALP treatment leads to an up-regulation of *Sp5* in the upper and lower body column, together with a down-regulation in head tissue. We favor a model where very high levels of nuclear β -catenin, as observed at the apical tip of untreated animals, leads to the expression of a negative regulator of *Sp5* expression and thus to the exclusion of *Sp5* from the head organizer.

In particular in Fig 3d, the authors need to show whether Sp5 is expressed in the very apical – ie can b-catenin drive Sp5 into the apical region even in the presence of wnt3 expression – a higher magnification of the apical domain is needed to assess this important data. Thus the authors show nicely that Sp5 is a target of nbcatenin – but its not at all clear what this means for how the spatial territories are established and hence how the head domain is established. Does Bcatenin activate wnt3 in just the apical domain or is the model that Wnt3 is activated more broadly and then restricted by Sp5. This is an important component of an understanding of how Sp5 is functioning to limit the head domain. This could be determined by examining wnt3 expression in b-catenin and Sp5 double perturbations.

M.V. et al: This comment is partially overlapping with the Reviewer's first comment. *Sp5* is not expressed at the very apical tip as shown in **Fig. 1g, Supplementary Figure 4 (red arrowheads)**. As mentioned above we believe that in the presence of high levels of *Wnt3*, a negative regulator of *Sp5* expression is produced. Now in animals knocked-down for *Sp5* and maintained in physiological conditions, we never detected a broad domain of *Wnt3* expression, likely as *Sp5* regulation is highly dynamic (see the variability of the expression patterns in **Supplementary Figure 4** and the NEW section in the discussion). As a consequence, *Sp5* transcript levels cannot be maintained low for long periods of time upon siRNA electroporation.

We managed to produce broader domains of *Wnt3* expression along the body column of animals where β -catenin is constitutively maintained active (upon ALP treatment) and knocked-down for *Sp5* (see **Figure 3c**). This diffuse up-regulation of *Wnt3* in the body column becomes visible, at least transiently for few days, as *Sp5* is less responsive to ALP treatment than *Wnt3* (so less up-regulated see qPCR data in **Supplementary Fig. 14b**). In these conditions we clearly modify the balance between activator and repressor. Therefore, we favor a model where *Wnt3* expression is activated broadly and subsequently restricted by *Sp5*.

Also the authors make the case in the intro that the HI should be graded along the apical basal axis, but I'm not sure why this is needed for the model, but this is in part because

their model is not clear? But if this is the case the authors need to explain why, and then need to provide better case that this is how Sp5 is expressed. The insitu expression doesn't look at all graded (Fig 1g and 3d – and 3d in particular looks completely uniform apart from the very base) and the quantitative profile should have some statistical analysis to show that expression is indeed graded. Also, what is staining at the very base – is this background?. Finally, the authors should really show that the protein levels are graded to make a case for a graded profile. But in the end its not at all clear anyway why a graded expression is needed to support the HI model.

M.V. et al: Several transplantation experiments demonstrated that the activity of the head inhibitor and activator is graded along the body axis (*Webster 1966 J. Embryol. Exp. Morph; Takano and Sugiyama 1983, J. Embryol. Exp. Morph.*). Thus one of the five criteria that we initially fixed to identify putative head inhibitor candidate genes was an apical to basal graded activity (**Fig. 1b**). However, given that we selected candidate genes based on RNA-seq data and that the *Wnt3* RNA-seq profile resembles a graded expression (**Fig. 1e-f**), we assumed that a graded expression might be equivalent to a graded activity, which, of course might not be valid.

We agree that it might be difficult to appreciate an apical-to-basal graded expression of *Sp5* in intact polyps. We changed the terminology to *Sp5* is predominantly expressed in the head. By contrast, one can appreciate a graded *Sp5* expression in regenerating conditions (see **Fig. 1g**, and also **Supplementary Fig. 4**). Still, we noted some slight variations of the *Sp5* expression pattern from animal to animal. This suggested to us that the expression of *Sp5* might be oscillating, which would further prevent the formation of an apical-to-basal graded expression pattern in intact animals. We do not consider the staining at the base as an unspecific signal as the RNA-seq data shows that *Sp5* is also weakly expressed in the lowest region of intact animals (**Fig. 1f**). Several attempts to produce an antibody against *Sp5* failed. Therefore, we can currently not test whether the *Sp5* protein levels are graded along the body axis.

Given the model that they develop, I don't understand why Wnt3 expression (and the supernumerary heads) arise in the basal regions in Sp5 RNAi when Sp5 is not especially expressed in this region (according to their data in Fig 1). Why are there not supernumerary heads in the apical domain in non-regenerating polyps. And then why does this differ in the regeneration assay, where heads are now all apical. These supernumerary head phenotype are a significant finding of the manuscript but the authors have not explained this phenomena with their model.

M.V. et al: The qRNA-seq data demonstrate that *Sp5* is indeed expressed in the body column of intact animals (**Fig. 1f**) and we discuss above the variability of *Sp5* expression in the body column of intact animals. As detailed above in the reply to Reviewer-2. In intact animals, formation of ectopic axis first occurs in the budding region, which is a developmentally-competent zone and a region of low *Sp5* expression, so easier to silence. In this region multiple heads differentiate (see white arrows of **Supplementary**

Fig. 6a). With increasing numbers of electroporations, ectopic axes also develop in the upper body column, however they remain headless due to the high expression of *Sp5* in this region (see red arrow of **Supplementary Fig. 6a**). In contrast, in a head-regenerating *Sp5*(RNAi) animal the level of *Sp5* is much lower compared to the head of an intact polyp, which therefore enables the formation of supernumerary axes in the apical region. Finally during reaggregation, knocking-down *Sp5* leads to a supernumerary phenotype with an increased number of *Wnt3* spots when *Wnt3*/ β -catenin signaling activity is high (ALP treatment). Again the critical point is the balance between the activator and inhibitor components.

In summary, I think that the authors have well justified the regulatory loops shown in figure 4j and that this is conserved. However they have failed to provide of model of what this means for how the head is organized during homeostasis and regeneration. I think the manuscript would be greatly strengthened by providing a model that explains how the head is restricted to the apical pole and how various perturbations, e.g. loss of Sp5, perturbation of bcatenin, regeneration leads to the observed heads phenotypes shown.

M.V. et al: Our main finding is that *Sp5* acts as a negative regulator of *Wnt*/ β -catenin signaling and throughout the manuscript we clearly provide convincing evidence that *Sp5* acts as a transcriptional repressor of *Wnt3* expression and thus restricts head organizer activity in *Hydra*. We refined our model that we now present in main Figure 7, that we hope to further dissect in the future with investigations based on sensors of *Sp5* activity.

The final sentence of the manuscript 181-185 completely misrepresents the significance of the paper – the significance should be the model and how it relates to previous theories that explain the head formation. It has nothing to do with cancer biology – this is a throwaway sentence that weakens the manuscript.

M.V. et al: We removed this sentence from the new version of our manuscript.

Minor comments

lines 90-91– does not show that Sp5 directly regulates the wnt3 promoter – as loss of DBD from Sp5 could activate or repress an intermediate gene. Experiments of line 94-96 do provide this evidence.

M.V. et al: We demonstrate in **Figure 4c** that *HySp5* directly regulates the *HyWnt3* promoter as the repressing effect on the *Wnt3* promoter was lost when using a reporter construct lacking the two border regions of the *Wnt3* repressor element. As a complement we provide cell culture and ChIP-qPCR data, showing that zebrafish *Sp5* directly regulates the zebrafish *Wnt3* promoter (**Fig. 4d-f**) and furthermore the ChIP-seq data demonstrate that both *Hydra* and zebrafish *Sp5* likely regulate the human *Wnt3*

promoter through the different binding sites we have identified (**Supplementary Fig. 16c**).

What is the significance of the expression in gESCs

M.V. et al: Gastrodermal epithelial cells are the epithelial cells from the inner layer of the animals, known in the field to carry morphogenetic processes. *Sp5* is mainly expressed in these cells, suggesting that *Sp5* can regulate morphogenesis in *Hydra*.

Reviewer #4

*The authors provide strong experimental evidence for the evolutionarily conserved nature of a *Sp5* and *Wnt/β-catenin* interaction across metazoa, factors previously known to interact in vertebrates. They do this by testing *Hydra Sp5/Wnt* interaction in mammalian cell culture and in the zebrafish embryo. Using these experimental systems they provide strong evidence for autocatalytic activation of *HySp5*, direct regulation of *Hysp5* by β -catenin/TCF, direct modulation of *wnt* through *Hysp5* and *HySp5* interaction with β -catenin and TCF-1. These results are compelling even though these interactions and a graded distribution of the *Sp5* gene product are not shown in the context of the animal. Within *Hydra* they seem to demonstrate *Sp5* function by performing RNAi.*

This work could be interesting for publication in Nature communications but there are a number of edits to the manuscript and clarifications necessary that I would like to see.

*The authors set out to identify the hypothesized diffuse head inhibitor. They did this by screening for genes that are expressed in a graded manner. Even though this approach is valuable it appears that a graded distribution of such an inhibitor may not necessitate graded expression. The identified factor is a transcription factor making it an unlikely candidate for the inhibitor as proposed in the reaction-diffusion system by the Gierer-Meinhardt as the authors acknowledge. Even though a key feature is missing they call it the head inhibitor in the sense of the original model? I do not think this is justified. I would rather suggest that the authors report the findings and discuss them in the light of the existing models for *Hydra* patterning. They could do so even though they originally set out to identify a diffusible factor. Could a conclusion be that patterning in *Hydra* does not involve diffusion and one has to rethink the model? I think that the statements made in the introduction and in the concluding paragraph (l. 26, l. 174) are therefore too strong and suggest that they are adjusted accordingly.*

M.V. et al: This is a very interesting comment. Rand et al demonstrated for the first time in 1926 that *Hydra* produces a head inhibitor. Since then many attempts were taken to identify this inhibitor, however without any success. A possible reason for the failure is likely that all studies focused on the identification of a diffusible substance. However,

throughout the literature there is no evidence that the head inhibitor is indeed diffusible. Furthermore, when Meinhardt and Gierer published their famous theory of biological pattern formation in 1972, transcription factors were not discovered yet. The first transcription factor ever discovered was the simian viral repressor SV40 T antigen in 1982 (*Saragosti et al., J Mol Biol 1982*), followed by the discovery of the first human transcription factor, SP1, in 1985 (*Jones and Tjian, Nature 1985*).

Over the last years it also turned out that the Meinhardt-Gierer model is too constrained and that “*realistic reaction-diffusion systems are based on mechanisms that are fundamentally different from the concepts of short-range activation and long-range inhibition based on differential diffusivity*” (Marcon et al. 2016). Thus, in the screen we designed to identify putative head inhibitor candidate genes, **we did not take diffusion into account**. We identified Sp5 as a key head inhibitory component of the freshwater polyp *Hydra* that is up to date the most promising candidate for the head inhibitor. As other scientists interested in this model (*Marcon et al., eLIFE 2016; Diego et al., Phys. Rev 2018*), we do believe that the Meinhardt-Gierer model needs to be revised. Nevertheless, we can currently not exclude that Sp5 might also regulate the expression of a diffusible inhibitor. We have therefore revised the introduction and the discussion accordingly.

Specific comments:

I. 1: Change title to read: evolutionarily conserved

M.V. et al: We changed the title.

I. 16: Suggested change: Polyps of the cnidarian Hydra maintain..

M.V. et al: We followed this advice.

I. 17: Suggested change: ..., apically the head organizer and basally the foot organizer.

M.V. et al: We followed this advice.

I. 19: This needs rewording: .. through the reactivation underneath the amputation plane of the proper organizer,

M.V. et al: We changed this sentence.

I. 21: Here we characterize.. , we identify... (present tense?).

M.V. et al: We followed this advice.

I. 26: Head Inhibitor, change to lower case.

M.V. et al: We made this change.

I. 46: The data sets mined (Fig. 1 c(2), c(3), e, f) have been previously discussed by the authors but raw data are not accessible and are not released in the course of this present study. I find this problematic in regards to reproducibility and especially since the foundation of the present study are publicly available data, publicly available data are used to identify Sp5 binding sites in mouse ESCs and also for comparison with genomic occupancies of β -catenin and Sp5 in mammalian cells.

M.V. et al: We agree with the reviewer that this is problematic in terms of reproducibility. We are therefore currently preparing a manuscript with a full release of the raw data. The RNA-sequencing data can already be accessed using the following link: https://hydratlas.unige.ch/blast/blast_link.cgi

I. 47: add: after 24h of regeneration.

M.V. et al: We added this information.

I. 49: this needs re-wording: "unique HI candidate"? It appears Sp5 has some features previously postulated for the head inhibitor.

M.V. et al: We changed this sentence.

I. 53: Is there are strong signal in the tentacles as well? The in situ hybridizations suggest so.

M.V. et al: Yes, we usually observe a Sp5 staining in the tentacles. However, cells populating the tentacles are considered as poorly active in terms of transcription. Therefore, this signal might correspond to Sp5 transcripts that remain stable in these cells.

I. 55-56: Fig. 1f,g. It appears there is a strong in situ signal in the regenerating foot during the first 24h and that disappears at time point 48h at which a foot has been regenerated? Do these results contradict the gene expression data. Are the authors considering this an unspecific signal?

M.V. et al: Sp5, is expressed in head- and foot-regenerating tips, however the expression is only sustained in the head regenerating part. This does not contradict with the RNA-seq data as we also obtained an up-regulation for Sp5 in the foot regenerating condition (see orange line in **Fig. 1f**). We do not consider the up-regulation of Sp5 in the regenerating foot as an unspecific signal as injury can trigger an initial up-regulation of

genes at both wound sites and many are later sustained at only one side (*Wenger et al., 2014 Seminars in Immunology*). To test a putative role of *Sp5* during foot regeneration we added new data and we can now demonstrate that all *Sp5*(RNAi) animals regenerate their foot normally (**Supplementary Fig. 7d**). Thus the up-regulation of *Sp5* in the foot is injury-related without having a role in head inhibition.

p. 2. l. 57. "An analysis of cell-type expression by RNA-seq18 showed that both Sp5 and Wnt3 are predominantly expressed in the gastrodermal epithelial stem cells."

The authors are referring to another dataset that was first presented in 2016. The data have also been used in several studies the raw sequencing data are to date not available. It appears that both the regeneration and tissue specific data sets were created for Hydra vulgaris (strain Jussy)? This should be stated in the methods section.

M.V. et al: The *Jussy* strain (*Hydra vulgaris* species) was used to generate the homeostatic spatial and regeneration data sets while the *AEP* strain (also *Hydra vulgaris* species) was used to generate the cell-type specific datasets. We added this information to the Supplementary Materials and Methods section.

l. 63: change: "not from" to "in case of".

M.V. et al: We prefer to say that multiple heads differentiate when located in the basal half but not from the upper half.

l. 66: The terminology should be changed throughout. I do not think "clustered apical cells" is the right way to describe this. Maybe: "Cells of the hypostomal tip", "a distinct population of cells in the apical tip" or "a distinct cluster/group of cells".

M.V. et al: We revised the text accordingly.

l. 66: This phenotype is truly compelling. The authors should provide explicit detail on the electroporation strategy in the methods section or in the supplement including media used, incubation times etc. Did all animals survive this relatively harsh treatment? In previous reports electroporation resulted in a patchy knock down. Is this maybe the reason why new axes form and why they do not find ectopic tentacle formation as in case of Alsterpaulone treatment? I would like to see an in situ hybridization of Sp5 in RNAi treated animals. Are there patches free of Sp5 transcript? Did the authors check via qPCR if Sp5 message was significantly depleted in the RNAi treated animals?

M.V. et al: We added additional information about the RNAi procedure to the Materials and Methods section. The electroporation conditions that we describe are optimized so that all animals survive the electroporation. Over the last months, we generated more *in vivo* data to demonstrate that *Sp5* prevents ectopic head formation by repressing the expression of *Wnt3*. This data set is now presented in the new main **Figure 3**. Indeed,

treating *Hydra* with ALP leads to ectopic tentacle formation and the laboratory of Thomas Holstein showed that these animals subsequently develop ectopic heads (*Guder et al., 2005 Development*).

When we tested the expression of *Sp5* after knocking down *Sp5*, we saw an up-regulation of *Sp5*. We believe that these results can be explained in different ways. The knockdown of *Sp5* leads to an immediate de-repression of *Wnt3*, which might in turn trigger the up-regulation of *Sp5*. Another scenario could be that *Sp5* has a repressing effect on its own promoter as described for human SP5 in embryonic stem cells (*Huggins et al., 2017 Nature Commun*). We tried hard to generate a transgenic *HySp5* reporter line that could have been used to follow the knockdown of *Sp5* in live animals (also see response to Reviewer 1). However, technical limitations currently prevent us from generating such a line. We have modified the model and also the discussion accordingly.

I. 80-82: Do the authors find that axis formation precedes wnt expression? Please clarify. The formation of a new axis in case of budding is indicated by wnt expression even prior to visible evagination (Hobmayer et al. 2000, Fig. 2i). Is this different in Sp5 RNAi animals that form ectopic axes? Maybe the transgenic animals expressing HyWnt3-2149:GFP could be used to combine Sp5 in situ hybridization with GFP immunohistochemistry to further elucidate the interaction.

M.V. et al: Hobmayer et al. showed an up-regulation of *TCF* and *β-catenin* expression prior to evagination. However, the animal presented in Fig. 2i shows a spotty expression of *Wnt3* at early budding stage. We performed an extensive time course experiment to monitor *Wnt3* expression immediately after *Sp5* knockdown and 72 hours after the first RNAi, we could observe the emergence of *Wnt3* expression in clustered cells (see animal 1 of **Supplementary Fig. 12c**), similarly to what has been shown in Fig. 2i of Hobmayer et al. 2000. Thus, *Wnt3* expression does not seem to be globally up-regulated in these conditions (*Sp5*(RNAi) animals that form ectopic axis), or at least only transiently as observed after ALP treatment and *Sp5* RNAi (see **Figure 3c**).

I. 86: “As the HyWnt3-2149:Luc construct..”? The authors should briefly introduce their approach or rephrase. Maybe add “..using a luciferase reporter assay” at the end of the preceding sentence.

M.V. et al: We added a sentence clarifying that we performed luciferase reporter assays.

I. 88 – I. 100. These are compelling findings.

I. 111: I suggest to change to “axial patterning”.

M.V. et al: We rephrased this sentence.

I. 114: consistent

M.V. et al: We changed this sentence.

I. 157: It feels this needs rephrasing. Change to: “These predicted consensus matrixes allowed for the identification of putative Sp5-binding...”.

M.V. et al: We changed this sentence.

I. 174: The authors claim that they have identified a predicted head inhibitor, a factor for which a diffusive nature has been postulated. I do not think this claim is justified since there is still the possibility of a such a factor with a role in head inhibition. The results are interesting in general and the authors clearly accumulated strong evidence for an evolutionarily conserved interaction that has an important role in Hydra patterning. This alone is worth reporting.

M.V. et al: We agree with the reviewer that a diffusible inhibitor might still exist. We added a short paragraph to the discussion.

I. 179: I suggest to re-word. It appears that the Sp5 interaction in Hydra development and its evolutionarily conserved nature are previously uncharacterized. The interaction itself appears phylogenetically old and has been characterized in vertebrates.

M.V. et al: We removed this sentence completely from the new version of our manuscript.

Figure & Legends

Fig. 4 d): The arrows from gray boxes seem to directly point to FIMO box bypassing green, yellow and purple boxes. I do not think this what the authors intend to show.

M.V. et al: We changed the arrows so that they do not directly point to the FIMO box.

Material and Methods

Table 2 (a): spelling: pGEM-T-Easy

M.V. et al: Thanks for pointing this out.

I. 283: 3,666ng were used? Please provide a concentration or a volume. What were the plasmids used? What was the concentration of the cells?

M.V. et al: 3,666ng were transfected. We added the cell concentration, the volume of the medium and the plasmids used.

I. 309: Please provide cell concentration and volume.

M.V. et al: We added this information.

Extended data

Fig. 3: What resources were mined to get Sp/Klf related genes from sponges? Missing genes could be just a function of the reference that was searched? Are there other sponge references that could be consulted? It does not appear to be a thorough phylogenetic analysis of this gene family. The figure title seems therefore too strong and should be changed. Maybe just provide a statement on similarity and grouping of HySp5 when compared to the available sequences. The last sentences of the figure legend need to be changed accordingly.

M.V. et al: The tree shown in supplementary Figure 3 is one of the multiple trees we have done, some of them with 100 different sequences, some with MAFFT alignment, others with Muscle Align. The tree we show was trimmed to 56 sequences as the most parsimonious and the sorting of the three Sp families confirmed by the other trees. All sequences were retrieved on Uniprot.org or NCBI from species carefully selected for two reasons: (1) they represent the major super phyla in metazoans and non metazoans as indicated in the legend and (2) they have a rather well assembled genomic sequences or complete transcriptomics. A similar choice is made by all scientists who analyze the origins and evolution of gene families, in fact Capsaspora and choanoflagellates are very informative to trace gene families that originated prior to metazoans. Among porifers, *Amphimedon queenslandica* is the only species so far to fulfill these criteria. As naming of these sequences was confusing, we decide to sort the Sp and KLF sequences, and use WT1 sequences as outgroup. From these multiple trees, we never found any non-vertebrate sequence grouping with the Sp1/2/3/4 super family, and we found two superfamilies that have representatives in both bilaterians and cnidarians: the superfamily Sp6/7/8/9 and the Sp5 family. The basal position of the porifera sequence within the Sp sequences strongly suggests that this sequence represent a single gene family that precedes its duplication in two eumetazoan families. We do not claim that other clades within the Porifera phylum do not already express orthologs of the Sp6/7/8/9 and Sp5 family, but this probability seems rather low.

The sequence from Amphimedon was retrieved as all sequences from Uniprot.org, completed with related sequences from NCBI; it was initially misnamed "Gli3" as NCBI sequence (gil761911773|reflXP_011405146.1| PREDICTED: transcriptional activator GLI3-like [Amphimedon queenslandica]) renamed as Btdl (Buttonhead-like) at Uniprot.-org (A0A1X7UG71_AMPQE Uncharacterized protein OS=Amphimedon queenslandica

PE=4 SV=1). Misnaming sequences before the evolutionary picture is made visible, is very common.

- replace “affiliated to” with “of the”

M.V. et al: We changed this sentence.

- replace “not affiliated” with “does not group with”?

M.V. et al: We changed this sentence.

- in the legend replace “porifers” with “poriferans”

M.V. et al: We did this change.

- in the figure legend (*Capow, filasterean*)

M.V. et al: We did this change.

Fig. 7: update to read “(see supplementary material and methods)”.

M.V. et al: Extended Data Figure 7 is now **Supplementary Figure 13** and we added this information.

Fig. 8: add ““(see supplementary material and methods)”.

M.V. et al: Extended Data Figure 8 is now **Supplementary Figure 14a**. However, we do not provide any information regarding this figure in the supplementary materials and methods section.

Supplementary Information

Hydra genome assembly:

The authors should provide details on sequencing depth, sequencing effort and general stats such as N50, number of scaffolds, contigs. This is valuable information. Did Hydra viridissima animals contain symbionts?

M.V. et al: We added this information to the Supplementary Materials and Methods section. *Hydra viridissima* contains symbionts.

Reviewers' Comments:

Reviewer #1:

Remarks to the Author:

The authors have significantly improved the manuscript and have satisfactorily addressed all of my concerns.

There are two remaining minor points I would like to see addressed in the final manuscript:

Figure 6i shows Venn diagrams that summarize the number of genes directly regulated by hydra and zebrafish Sp5. The gene IDs are buried in a Supplemental Table. The authors should provide the gene names for these relatively short lists of genes, and for simplicity sake, provide a reference to this data in the text or in the figure legend. Also, a gene ontology analysis of these genes may be informative.

Since this manuscript addresses the evolutionarily conserved function of Wnt signaling in the polarization of the primary axis, it would be worthwhile to mention that the oral end of cnidarians corresponds to the posterior end of bilaterians. This is a minor point, however, a simple statement addressing this point somewhere in the text may offset confusion among those who are less well-versed in metazoan primary axis formation and associate Wnt signaling activity with the posterior end of an organism.

Reviewer #2:

Remarks to the Author:

The authors have thoughtfully, thoroughly and carefully addressed our concerns.

Reviewer #3:

Remarks to the Author:

The authors have appropriately addressed all of the concerns from the first round of reviews and made significant improvements to the manuscript. This paper will be a significant contribution and should find a wide audience.

Reviewer #4:

Remarks to the Author:

The changes made and additional experiments significantly improved the manuscript and I'm in support of publication.

The postulated negative regulator that excludes Sp5 from the hypostomal tip could be added to the module presented in Fig. 7.

Reviewer #1 (Remarks to the Author):

The authors have significantly improved the manuscript and have satisfactorily addressed all of my concerns.

There are two remaining minor points I would like to see addressed in the final manuscript:

Figure 6i shows Venn diagrams that summarize the number of genes directly regulated by hydra and zebrafish Sp5. The gene IDs are buried in a Supplemental Table. The authors should provide the gene names for these relatively short lists of genes, and for simplicity sake, provide a reference to this data in the text or in the figure legend. Also, a gene ontology analysis of these genes may be informative.

M.V. et al: We agree with the reviewer and we now show the gene names in Supplementary Figure 17 and Supplementary Data 3 as well as provide a reference to this data in the text. In addition, we performed a GO analysis that is presented in Supplementary Figure 17.

Since this manuscript addresses the evolutionarily conserved function of Wnt signaling in the polarization of the primary axis, it would be worthwhile to mention that the oral end of cnidarians corresponds to the posterior end of bilaterians. This is a minor point, however, a simple statement addressing this point somewhere in the text may offset confusion among those who are less well-versed in metazoan primary axis formation and associate Wnt signaling activity with the posterior end of an organism.

M.V. et al: We added a sentence to the text, clarifying that it has been suggested the oral end of cnidarians corresponds to the posterior end in bilaterians.

We are grateful for the helpful and constructive comments of the reviewer that helped to improve the manuscript significantly.

--

Reviewer #2 (Remarks to the Author):

The authors have thoughtfully, thoroughly and carefully addressed our concerns.

M.V. et al: We thank the reviewer for the constructive comments and suggestions that helped to improve the manuscript.

--

Reviewer #3 (Remarks to the Author):

The authors have appropriately addressed all of the concerns from the first round of reviews and made significant improvements to the manuscript. This paper will be a

significant contribution and should find a wide audience.

M.V. et al: We thank the reviewer for the comments and suggestions that helped to improve the manuscript.

--

Reviewer #4 (Remarks to the Author):

The changes made and additional experiments significantly improved the manuscript and I'm in support of publication.

The postulated negative regulator that excludes Sp5 from the hypostomal tip could be added to the module presented in Fig. 7.

M.V. et al: For simplicity, we prefer to not show this putative negative regulator in Figure 7. We thank the reviewer for all the previously constructive comments and suggestions.